# Self-triggered thermoelectric nanoheterojunction for cancer catalytic and immunotherapy

Xue Yuan[1,3], Yong Kang [1,3], Jinrui Dong[1,3], Ruiyan Li[1], Jiamin Ye[1], Yueyue Fan[1], Jingwen Han[1], Junhui Yu[1], Guangjian Ni [1], Xiaoyuan Ji [1,2] ✉ & Dong Ming[1]

The exogenous excitation requirement and electron-hole recombination are the key elements limiting the application of catalytic therapies. Here a tumor microenvironment (TME)-specific self-triggered thermoelectric nanoheterojunction ($Bi_{0.5}Sb_{1.5}Te_3$/$CaO_2$ nanosheets, BST/$CaO_2$ NSs) with self-built-in electric field facilitated charge separation is fabricated. Upon exposure to TME, the $CaO_2$ coating undergoes rapid hydrolysis, releasing $Ca^{2+}$, $H_2O_2$, and heat. The resulting temperature difference on the BST NSs initiates a thermoelectric effect, driving reactive oxygen species production. $H_2O_2$ not only serves as a substrate supplement for ROS generation but also dysregulates $Ca^{2+}$ channels, preventing $Ca^{2+}$ efflux. This further exacerbates calcium overload-mediated therapy. Additionally, $Ca^{2+}$ promotes DC maturation and tumor antigen presentation, facilitating immunotherapy. It is worth noting that the $CaO_2$ NP coating hydrolyzes very slowly in normal cells, releasing $Ca^{2+}$ and $O_2$ without causing any adverse effects. Tumor-specific self-triggered thermoelectric nanoheterojunction combined catalytic therapy, ion interference therapy, and immunotherapy exhibit excellent antitumor performance in female mice.

Reactive oxygen species (ROS) are crucial regulators of intracellular reduction–oxidation (redox) homeostasis and play a significant role in tumorigenesis and progression[1-6]. However, ROS can also induce apoptosis in tumor cells, making them useful tools for cancer therapy[2,3,7-10]. To this end, researchers have developed the ROS surging strategy, referred to as catalytic therapy, which includes photocatalytic therapy and piezocatalytic therapy[1,11-15]. These therapies have been widely reported in diverse cancer treatments, with several studies highlighting their efficacy in inducing tumor cell apoptosis. Although photocatalytic therapy is capable of converting light energy into chemical energy, its practical application is limited by various factors[10,16-22]. Therefore, alternative approaches such as piezocatalytic therapy have been explored to address these limitations and improve cancer treatment outcomes. In photocatalysis, light irradiation is necessary to initiate the process. However, this requirement adds complexity and limits its effectiveness since traditional photocatalysts have limited access to visible light energy in living organisms due to the barrier function of skin and other biological tissues. This scarcity of extrinsic light energy results in a limitation of the photocatalytic process[17,22]. To meet biomedical requirements, the fast recombination of photoexcited electron-hole pairs in both the surface and bulk phases of photocatalysts needs to be minimized[11]. This has been a significant challenge for photocatalytic therapy. However, piezocatalytic therapy based on the piezoelectric effect offers an alternative approach. By generating a piezoelectric potential, this therapy can drive charge separation or transfer and trigger redox reactions[23-27]. Essentially, this converts mechanical energy into chemical energy, but it does require the use of an additional external force, such as an ultrasonic generator.

In recent years, research on thermoelectric catalysis has increased due to its potential applications in environmental remediation and

[1]Academy of Medical Engineering and Translational Medicine, Medical College, Tianjin University, 300072 Tianjin, China. [2]Medical College, Linyi University, 276000 Linyi, China. [3]These authors contributed equally: Xue Yuan, Yong Kang, Jinrui Dong. ✉e-mail: jixiaoyuan@tju.edu.cn

energy supplementation. Thermoelectric catalysis involves combining the thermoelectric effect with chemical redox reactions to produce pyro-generated negative and positive charges for chemical oxidation–reduction reactions[28–32]. Unlike photocatalysis and piezocatalysis, temperature fluctuation is the trigger for this process[30,33–36]. Temperature difference causes slight spatial movements of atoms in the crystal structure, leading to polarization changes within pyroelectricity and induced thermoelectric charges on the surfaces of thermoelectric catalysts[30,34]. The generated self-built-in electric field between the opposite surfaces of thermoelectric materials can retard charge recombination, ensuring high ROS production and catalytic activity[37,38]. However, there have been relatively few investigations into ROS generation from thermoelectric catalysts. The high Seebeck coefficient of a thermoelectric material plays a critical role in efficient thermocatalytic performance, as it determines the thermoelectric voltage under a particular temperature difference[37,38]. While some biomedical applications of the thermoelectric effect have been reported, particularly in photothermal conversion of laser irradiation, this approach suffers from limitations such as limited penetration into biological tissue and low catalytic efficiency[38–43]. Therefore, the development of self-triggered thermoelectric catalytic materials or systems that maintain the advantages of thermoelectric catalysis while avoiding these limitations holds great promise for clinical transformation.

To enable biomedical applications in vivo, it is essential to have thermoelectric materials with high conversion efficiency at low temperatures. $Bi_xSb_{2-x}Te_3$ NSs exhibit a high Seebeck coefficient, high electrical conductivity, low thermal conductivity, and high thermoelectric conversion, making them suitable for self-triggered thermoelectric cancer therapy[37,44]. $Bi_{0.5}Sb_{1.5}Te_3$ nanosheets (BST NSs) have been found to have the highest thermoelectric conversion efficiency at very low temperatures, making them ideal for biomedical applications in vivo[44]. Additionally, $CaO_2$ nanoparticles serve as a reservoir of calcium ions ($Ca^{2+}$) and hydrogen peroxide ($H_2O_2$) and have shown promising results in calcium overload-mediated therapy[45–47]. However, $CaO_2$ NPs also have the potential to act as an in vivo switch for self-triggering thermoelectric dynamic therapy due to their tumor microenvironment (low pH)-specific water liberation thermal effect, which has not yet been explored.

In this work, we present a self-triggered thermoelectric nanoheterojunction for enhanced tumor catalytic/immunotherapy. The researchers synthesized high-performance nanoscale thermoelectric biomaterials, BST NSs, with excellent thermoelectric conversion capability. TME-responsive $CaO_2$ NPs were grown in situ on the BST NSs to create BST/$CaO_2$ NSs. Once the BST/$CaO_2$ NS-based nanoheterojunction reached the tumor region through enhanced permeability and retention effect (EPR)-mediated passive targeting, the $CaO_2$ NP coating hydrolyzed rapidly under the stimulation of the acidic TME, liberating $Ca^{2+}$, $H_2O_2$, and a large amount of heat. The released heat induced a temperature difference on BST NSs, generating pyro-generated negative and positive charges for chemical oxidation–reduction reactions and ROS generation. The voltage inside BST NSs created a self-built-in electric field that retarded electron-hole recombination, ensuring corresponding catalytic activity and higher ROS production. The generated $H_2O_2$ provided substrate ($O_2$) supplementation for thermoelectric catalysis and ROS production, regulated $Ca^{2+}$ channels, and slowed $Ca^{2+}$ efflux. Released $Ca^{2+}$ mediates ion interference therapy, disrupting intracellular ion homeostasis and increasing the osmotic pressure of tumor cells, effectively killing cancer cells. Additionally, $Ca^{2+}$ improved DC maturation and presentation of tumor antigens, activating the immune response and mediating effective immunotherapy. Therefore, this self-triggered thermoelectric catalysis based on the BST/$CaO_2$ heterojunction combined catalytic therapy, ion interference therapy, and immunotherapy, exhibiting remarkable antitumor capability both in vitro and in vivo. This study provides an intelligent strategy for synthesizing self-triggered thermoelectric catalysts and an advanced strategy for improving the application scope and efficiency of catalytic therapy.

## Results

### Preparation and characterization of a self-triggered thermoelectric system

We applied a hydrothermal process to create the thermoelectric material BST using $Bi(NO_3)_3 \cdot 5H_2O$, $Na_2TeO_3$, and $SbCl_3$ as substrates and PVP as a blocker in an ethylene glycol solution (Fig. 1). The reaction system was heated for 8 h at 230 °C and then centrifuged the cooled reaction solution for 10 min at 1760×g to obtain BST NSs with uniform size distribution in the supernatant. As shown in Fig. 2a, e, both scanning electron microscopy (SEM) and transmission electron microscopy (TEM) images showed that the prepared BST NSs were orthohexagonal with an average size of 110 nm (Supplementary Fig. 1). Energy dispersive spectrometry (EDS) maps of BST NSs demonstrated that their chemical compositions included Bi (red), Sb (purple), and Te (green). $CaO_2$ NPs were synthesized using $CaCl_2$, ammonia, and $H_2O_2$ as substrates in an ethanol solution. Both SEM and TEM images of $CaO_2$ NPs showed that the synthesized $CaO_2$ NPs had a uniform morphology with an average size of 10 nm (Fig. 2b, f). EDS mapping of $CaO_2$ NPs also confirmed successful preparation (Fig. 2j, Supplementary Fig. 2). After confirming the successful preparation of BST NSs and $CaO_2$ NPs separately, the authors coated $CaO_2$ NPs on the surface of BST NSs via an in situ mineralization technique. After washing three times with pure water and centrifuging for 10 min at 4505×g, BST/$CaO_2$ NSs with a homogeneous size distribution were obtained. The morphology of BST remained orthohexagonal and possessed an average size of approximately 120 nm (Supplementary Fig. 3). The representative high-resolution scanning electron microscopy (HRSEM) and high-resolution transmission electron microscopy (HRTEM) images in Fig. 2d, h exhibit the rough $CaO_2$ NP coating and the main crystalline structure of BST. EDS mapping of BST/$CaO_2$ NSs was applied to demonstrate the successful $CaO_2$ NP coating. As shown in Fig. 2k, the representative elements of Bi (purple), Sb (yellow), and Te (orange) in BST and Ca (green) and O (red) in $CaO_2$ were exhibited in BST/$CaO_2$ NSs. Furthermore, the elements and zeta potential of BST/$CaO_2$ NSs were quantitatively analyzed. Additional information can be found in Supplementary Fig. 4 and Fig. 3a.

X-ray photoelectron spectroscopy (XPS) and X-ray diffraction (XRD) spectroscopy were employed to detect the chemical composition and crystal structure of BST NSs and BST/$CaO_2$ heterojunctions. In terms of XRD detection, the prepared orthohexagonal BST NSs exhibited high phase purity, which was evidenced by their XRD pattern (Fig. 3b). It is evident that all the XRD peaks were well matched with JCPDS card no. 00-049-1713 (corresponding to the orthohexagonal structure of BST nanocrystals, Supplementary Fig. 5). Furthermore, the XRD spectrum of the BST/$CaO_2$ heterojunction comprised the corresponding XRD peaks of orthohexagonally structured BST and $CaO_2$ nanocrystals (Supplementary Fig. 6), further certifying the successful synthesis of high-purity BST/$CaO_2$ NSs. In terms of XPS detection, the corresponding Te, Sb, and Bi elements were present in the XPS spectra of BST NSs, while the corresponding Te, Sb, and Bi elements from BST and Ca and O from $CaO_2$ were all present in the XPS spectra of BST/$CaO_2$ NSs (Fig. 3c). Additionally, the typical high-resolution XPS spectra of Sb 3$d$, Te 3$d$, Bi 4$f$, Ca 2$p$, and O 1$s$ from BST NSs and CaO2 NPs were analyzed in detail. Specifically, Bi 4$f_{5/2}$, Bi 4$f_{7/2}$, Sb 3$d_{3/2}$, Sb 3$d_{5/2}$, Te 3$d_{3/2}$, Te 3$d_{5/2}$, Ca 2$p_{1/2}$, Ca 2$p_{3/2}$, and O 1$s$ were observed in the high-resolution XPS spectra of BST/$CaO_2$ NSs (Fig. 3d). Overall, these results demonstrate the successful fabrication of the thermoelectric material BST NSs and the self-triggered thermoelectric system BST/$CaO_2$ NSs. Additionally, the loading capacity of $CaO_2$ NPs on BST/$CaO_2$ NSs was detected and calculated to be approximately 10 wt% by inductively coupled plasma source mass spectrometry (ICP/MS).

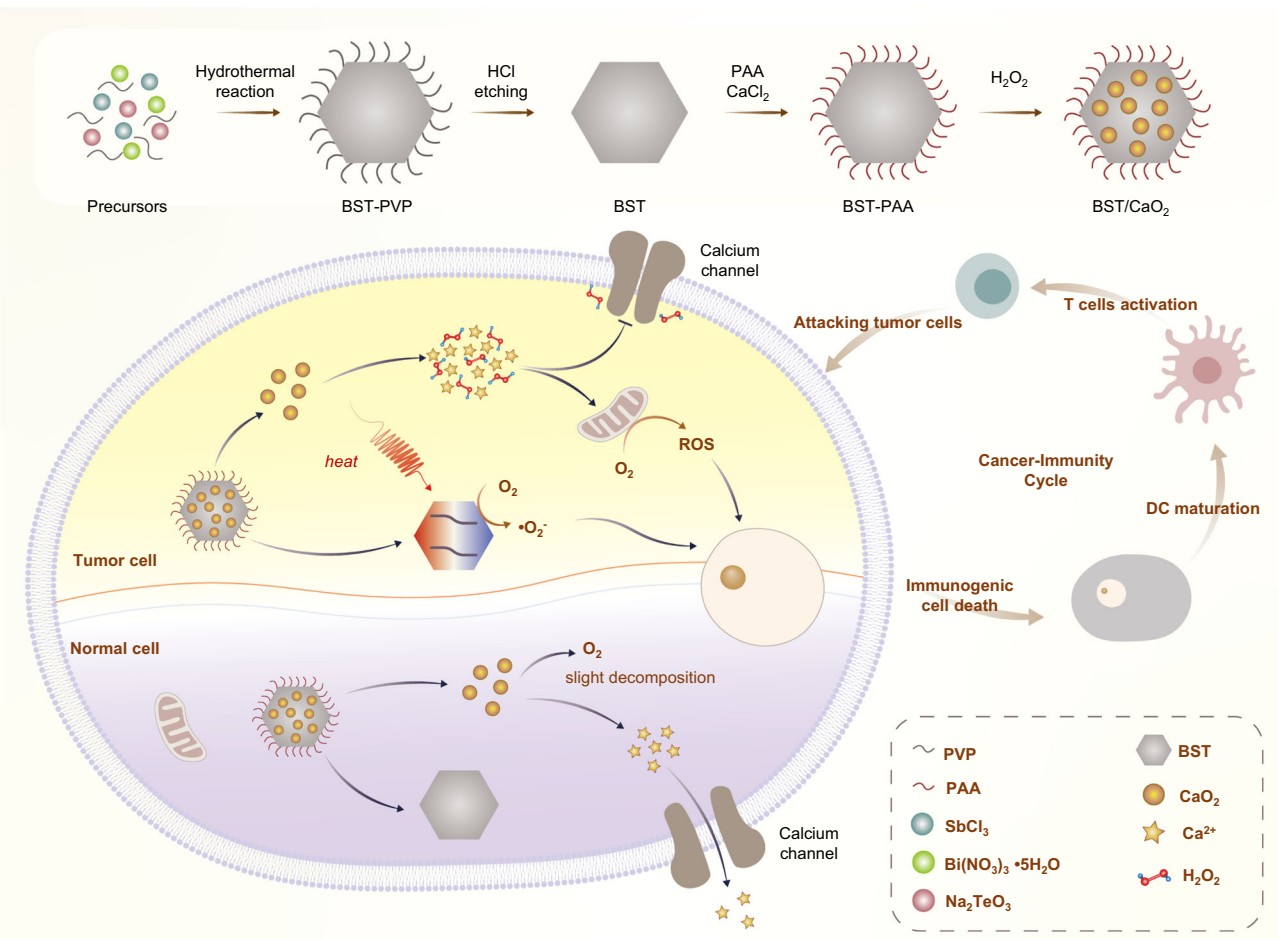

**Fig. 1 | Schematic diagram of the synthesis of BST/CaO₂ NSs and the mechanism of self-triggered thermoelectric dynamic therapy for cancer treatment.** BST Bi$_{0.5}$Sb$_{1.5}$Te$_3$, PVP polyvinylpyrrolidone, PAA polyacrylic acid, DC dendritic cells.

## Catalytic performance and mechanism of the self-triggered thermoelectric system

The use of highly oxidative reactive oxygen species (ROS) in catalytic therapy has proven to be an effective treatment for tumors[48–53]. In this study, the self-triggering thermoelectric catalytic performance of BST/CaO₂ NSs was investigated. Previously, BST NSs were found to be an efficient thermoelectric catalyst for cancer treatment when coupled with an external temperature difference stimulation system. However, this increased the complexity and imprecision of the operation[37]. The temperature difference of the thermoelectric catalyst is an indispensable condition for triggering the thermoelectric effect. Therefore, in this study, CaO₂ was assembled onto the surface of BST NSs in situ to create BST/CaO₂ NSs. The trigger (CaO₂) could only be activated by the low pH of the tumor microenvironment (TME), which triggered the thermoelectric effect. To test the theory of the low pH-specific self-triggered thermoelectric effect, the temperature change of BST NSs, CaO₂ NPs, and BST/CaO₂ NSs at different pH values was detected. It was observed that there was a rapid temperature rise when CaO₂ NPs or BST/CaO₂ NSs were placed in a low pH solution (pH 5.5), but there was no significant temperature fluctuation at neutral pH (pH 7.4) (Fig. 3e). Additionally, the ability of BST/CaO₂ NSs to generate ROS, mainly ·O₂⁻, was investigated using DPBF as an indicator (Fig. 3h). When BST NSs, CaO₂ NPs, and BST/CaO₂ NS suspensions were mixed with DPBF in PBS solution at pH 7.4 and room temperature (25 °C), there was almost no ROS generation (Fig. 3f and Supplementary Fig. 7). This phenomenon not only proved the mechanism of thermoelectric catalysis but also demonstrated the high biosafety of our prepared thermoelectric catalysts. However, when BST NSs were incubated in an

anisothermal environment with a temperature ranging from 25 to 45 °C, ROS generation was observed, triggering the thermoelectric effect in BST NSs due to the temperature difference (Supplementary Fig. 7). Finally, it was observed that the degradation of DPBF was more pronounced and rapid in the BST/CaO₂ NSs in the PBS (pH 5.5) group, further demonstrating the effectiveness of the self-triggered thermoelectric effect for generating ROS specifically in the low pH TME (Fig. 3g, i).

Low pH creates a unique microenvironment for tumor cells[54–57], and CaO₂ NPs are relatively stable in neutral environments but undergo violent hydrolysis reactions in low pH environments. This hydrolysis releases heat, triggering the thermoelectric catalytic switch of BST. To investigate the catalytic mechanism of BST NSs, first-principles calculations based on density functional theory (DFT) were performed using the projector augmented wave (PAW) approach. The optimized conformation of BST with the lowest total energy is illustrated in Fig. 4a–c. Sb atoms are uniformly doped into two Bi sublayers. The lattice parameters of BST are 8.49 Å and 8.49 Å with a total energy of −79.01 eV. Electronic structures were further investigated, as shown in Fig. 4d, e. BST NSs are an indirect band gap semiconductor with a gap of 0.825 eV using standard PBE functionals, and the work function is 4.64 eV. The differences between the CBM (or VBM) and the Fermi level are 0.415 and 0.410 eV, respectively. From the orbital-resolved electronic density of states plot, it reads that the CBM of BST is contributed by Te 5$p$, Sb 5$p$ and Bi 6$p$ together, while its VBM is almost contributed by Te 5$p$, meaning that O₂ reduction and ·O₂⁻ generation would occur at Te sites. The adsorption behavior of O₂ on BST was investigated to further confirm the above hypothesis. As

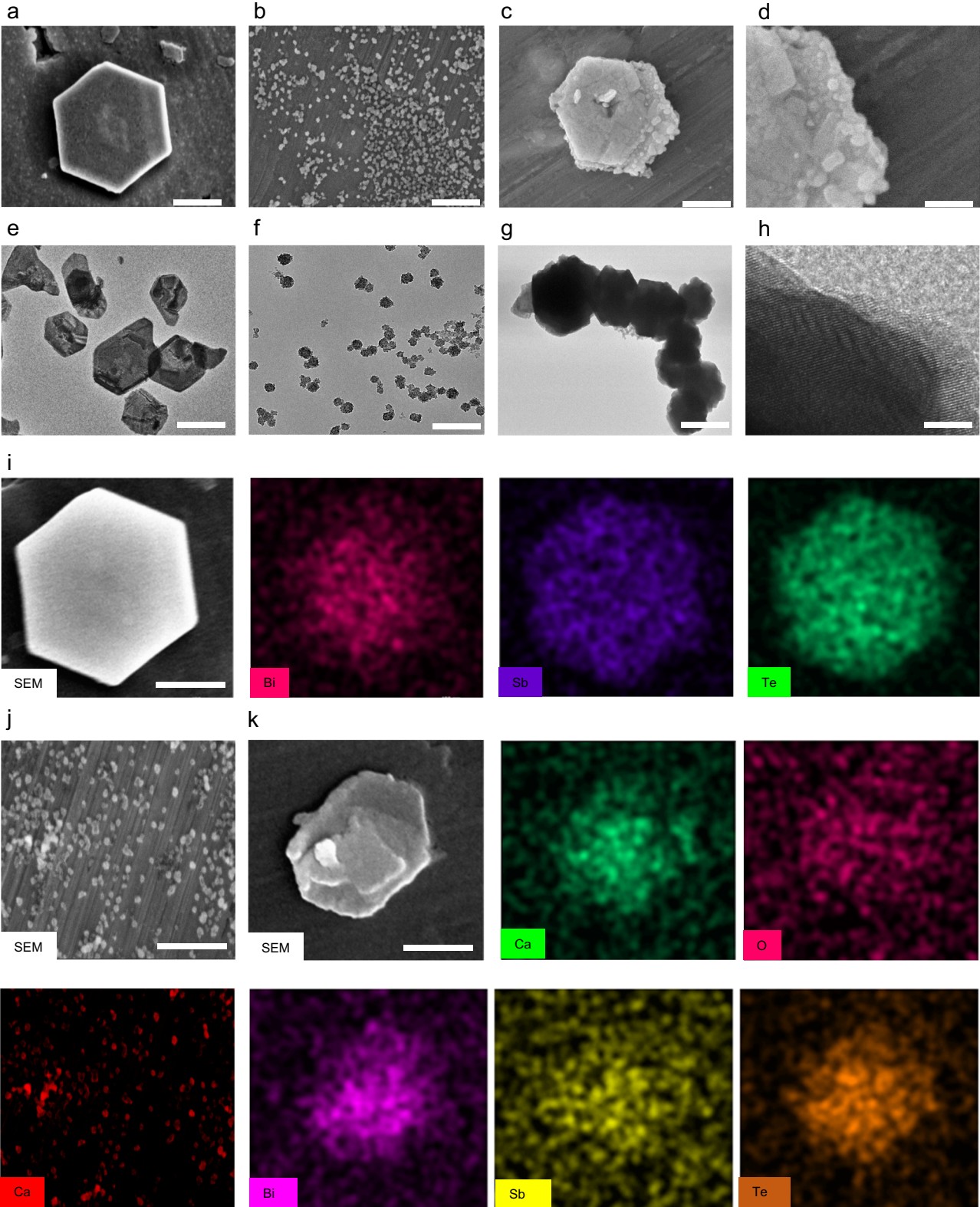

**Fig. 2 | Characterization of prepared BST NSs, CaO₂ NPs, and BST/CaO₂ NSs.**
**a** SEM images of BST NSs. Scale bar = 50 nm. **b** SEM images of CaO₂ NPs. Scale bar = 50 nm. **c**, **d** SEM images of BST/CaO₂ NSs. Scale bar = 50 nm for **c**, Scale bar = 10 nm for (**d**). **e** TEM images of BST NSs. Scale bar = 100 nm. **f** TEM images of CaO₂ NPs. Scale bar = 100 nm. **g**, **h** TEM images of BST/CaO₂ NSs. Scale bar = 100 nm for (**g**), Scale bar = 5 nm for (**h**). **i** SEM image and elemental mappings of BST NSs, including Bi, Sb, and Te elements. Scale bar = 100 nm. **j** SEM image and elemental mappings of CaO₂ NPs. Scale bar = 100 nm. **k** BST/CaO₂ NSs, including Bi, Sb, Te, Ca and O elements. Scale bar = 100 nm. For these morphology characterizations, three times each experiment was repeated independently with similar results.

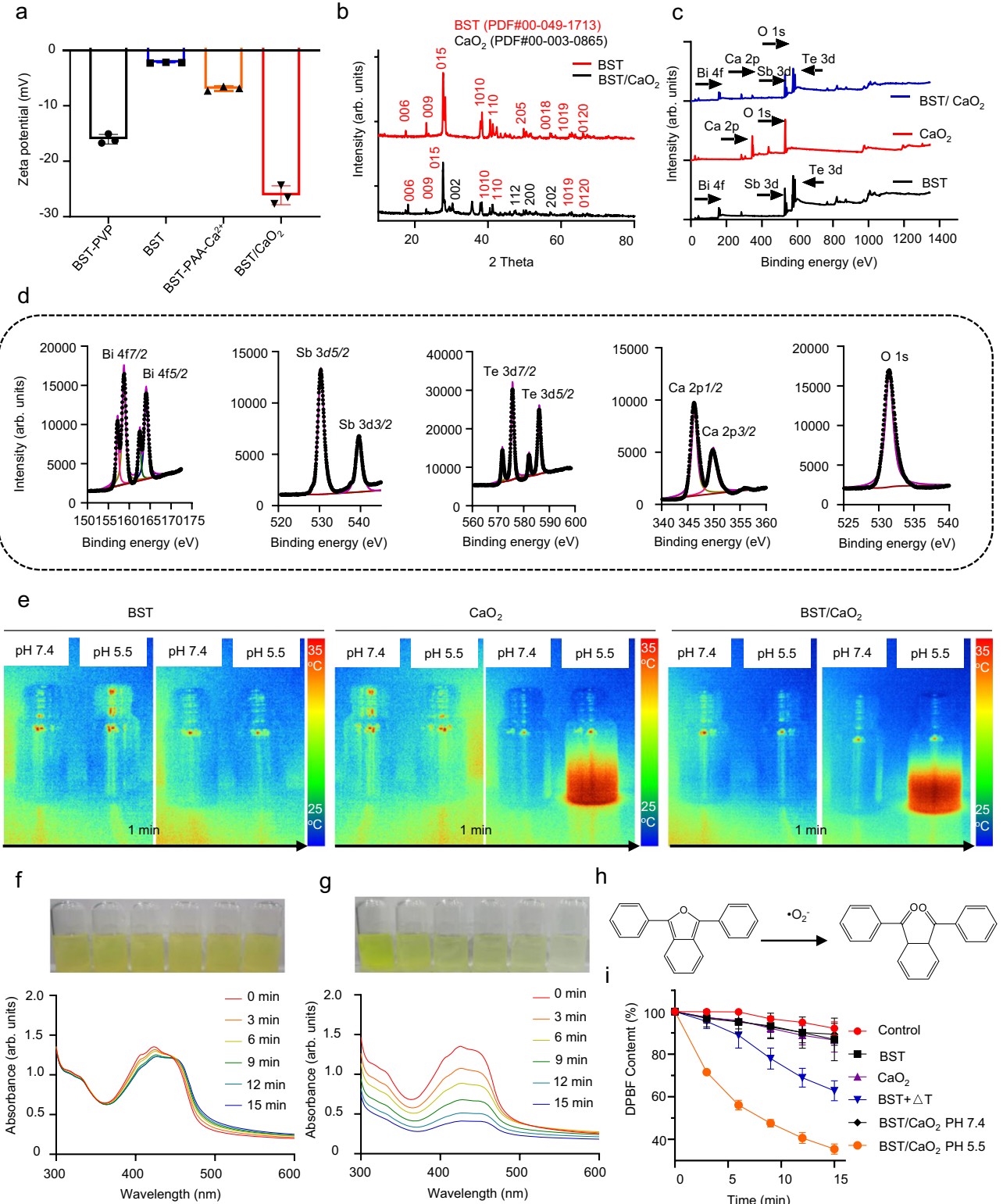

**Fig. 3 | Analysis of the catalytic performance of the self-triggered thermo-electric system. a** Zeta potential of ligand-free BST, BST-PAA-Ca²⁺, and BST/CaO₂ NSs. PAA means polyacrylic acid. Data are presented as the mean ± s.d. (*n* = 3 independent experiments). **b** X-ray diffraction (XRD) patterns of BST NSs and BST/CaO₂ NSs. **c** X-ray photoelectron spectroscopy (XPS) spectra of BST NSs, CaO₂ NPs, and BST/CaO₂ NSs. **d** High-resolution XPS spectra of BST/CaO₂ NSs (Bi 4f, Sb 3d, Te 3d, Ca 2p and O 1s). **e** The temperature change of BST NSs, CaO₂ NPs, and BST/CaO₂ NSs at different pH values. Degradation of DPBF by **f** BST and **g** BST/CaO₂ NSs at pH 5.5. **h** Reaction mechanism of DPBF detection ·O₂⁻. **i** Degradation of DPBF by different groups. Data are presented as the mean ± s.d. (*n* = 3 independent experiments).

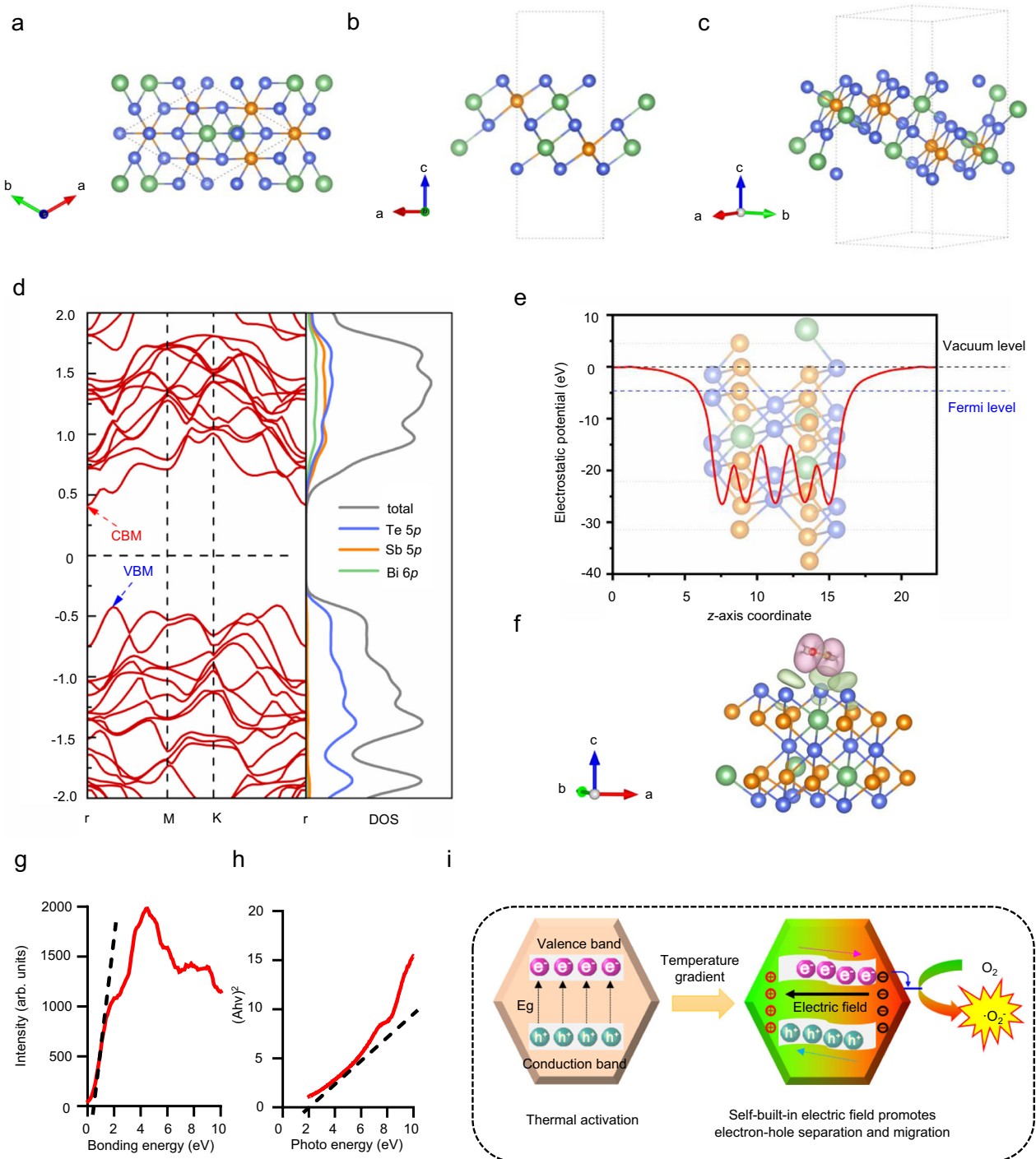

**Fig. 4 | Analysis of the catalytic mechanism of the self-triggered thermoelectric system. a** Top, **b** side and **c** perspective views of BST NSs. The blue, green and orange balls show the Te, Bi and Sb atoms, respectively. The dotted lines represent the unit cell. **d** Electronic band structure and orbital-resolved electronic density of states of BST using standard PBE functionals. The Fermi level is shifted to zero and indicated by the black dashed lines. **e** Plannar-averaged electrostatic potential (red solid line), vacuum level (black dashed line) and Fermi level (blue dashed line) of BST. The insets show the corresponding structures. The vacuum level was set to 0. **f** Optimized adsorption and charge transfer mode of $O_2$ on BST. Smooth pink and green shading show the electronic accumulation and depletion, respectively, with an isovalue of $1.5 \times 10^{-4}$ e/Å³. **g** Calculated valence band positions of BST/$CaO_2$ NSs. **h** Calculated band gaps of BST/$CaO_2$ NSs. **i** Analysis of the catalytic mechanism of the self-triggered thermoelectric system.

shown in Fig. 4f, the favorite adsorption energy between BST and $O_2$ is −0.13 eV, and this physical adsorption mediated by van der Waals forces allows $O_2$ to escape from BTS after electronic transfer. Furthermore, Bader's charge analysis shows that after $O_2$ adsorption, 0.1 e transferred from Te@BTS to $O_2$. Thus, the DFT calculations reveal that the electronic structure properties of BST make it an excellent catalyst

for $O_2$ reduction and $\cdot O_2^-$ generation. Furthermore, in Fig. 4g, it is evident that the XPS analysis showed the valence band (VB) of BST to be 0.5 eV. Additionally, according to Fig. 4h, the band gap (Eg) of BST was calculated to be 1.0 eV. Therefore, the conductive band (CB) of BST was determined to be −0.5 eV, which is more negative than the redox potential of $O_2/\cdot O_2^-$. The mechanism diagram of the

thermocatalytic production of ROS is illustrated in Fig. 4i. Specifically, when BST/CaO$_2$ NSs entered the tumor TME, the low pH promoted the hydrolysis of CaO$_2$ NPs, releasing a significant amount of heat. This heat-induced temperature difference triggered the thermocatalytic effect of BST, leading to electron-hole separation and transition to the CB of BST. Moreover, as the temperature difference was applied, it created a potential difference between the hot and cold ends of the thermoelectric catalyst, causing the negative charges to rush from the hot side to the cold side of the material. Due to the presence of the thermoelectric potential, the band energy increased at the negative potential side and decreased at the positive potential side. As a result, both the CB and VB were tilted across the material, with the CB being positioned very close to the redox potential required for producing ·O$_2^-$. Consequently, electrons from the CB of BST NSs could easily migrate to the solution, leading to the production of ·O$_2^-$.

### In vitro antitumor effects of the self-triggered thermoelectric system

After obtaining promising results from thermoelectric catalytic experiments, our next step was to conduct antitumor therapy in vitro. To ensure the safety of our treatment, we first evaluated the biocompatibility of three different types of nanoparticles—BST NSs, CaO$_2$ NPs, and BST/CaO$_2$ NSs—with normal cells such as human liver cells (HL-7702) and human embryonic kidney cells (HEK293). We used a CCK8 assay for this purpose. Our findings revealed that BST NSs did not exhibit any catalytic activity due to the stationary temperature and hence caused no significant cytotoxicity to normal cells (Supplementary Figs. 8 and 9). Similarly, CaO$_2$ NPs decomposed slowly into Ca$^{2+}$ and O$_2$ in a neutral environment without causing any significant heat generation or cytotoxicity to normal cells. For BST/CaO$_2$ NSs, since there was no triggering mechanism, the thermoelectric effect remained silent, and no cytotoxicity was observed in normal cells. Overall, all treatments showed excellent biosafety and biocompatibility with normal cells, indicating great potential for clinical transformation.

The cytotoxic effects of BST NSs, CaO$_2$ NPs, and BST/CaO$_2$ NSs on cancer cells were investigated. BST NSs showed minimal cytotoxic effects, while CaO$_2$ NPs exhibited specific cytotoxic effects on CT26 and TE1 cancer cells (as shown in Fig. 5a, b). This effect was likely due to two factors: Ca$^{2+}$ overload and H$_2$O$_2$ oxidative stress caused by the rapid hydrolysis of CaO$_2$ NPs in an acidic environment. When treated with BST/CaO$_2$ NSs, the highest cytotoxicity effect was observed, with more than 80% of tumor cells being killed at a dosage of 200 μg/mL. The good antitumor performance of BST/CaO$_2$ NSs can be attributed to the self-triggered thermoelectric effect generated by the hydrolysis of CaO$_2$ NPs within tumor cells. This effect efficiently catalyzed O$_2$ reduction for ·O$_2^-$ generation.

To validate the efficacy of calcium overload-mediated ion interference therapy, the intracellular concentrations of Ca$^{2+}$ were evaluated both qualitatively and quantitatively using confocal laser scanning microscopy (CLSM) and flow cytometry (FCM) with a Fluo-4 AM probe. The results depicted in Supplementary Fig. 10a indicate a significant increase in green fluorescence after treatment with CaO$_2$ NPs and BST/CaO$_2$ NSs, indicating successful endocytosis of these particles by tumor cells and hydrolysis of CaO$_2$ at low pH. Additionally, the quantitative data on intracellular Ca$^{2+}$ concentrations obtained via FCM analysis (Supplementary Fig. 10b) confirmed the high level of endocytosis observed for CaO$_2$ NPs and BST/CaO$_2$ NSs and the low pH-responsive hydrolysis of CaO$_2$ in tumor cells.

The ability of BST/CaO$_2$ NSs to induce tumor cell death is thought to be mainly due to the production of ROS induced by oxidative stress. This was evaluated using DCFH-DA as an ROS probe to measure intracellular ROS production levels. Under the influence of ROS, DCFH-DA can convert from nonfluorescent DCFH to fluorescent DCF, with the fluorescence intensity of DCF being directly proportional to the concentration of ROS. As illustrated in Fig. 5c, almost no fluorescent signal was observed in CT26 cells treated with BST NSs, which was similar to that of the control group. A weak fluorescent signal was detected in the CaO$_2$ group, likely due to H$_2$O$_2$ generated by the hydrolysis of CaO$_2$ NPs in an acidic environment. In contrast, tumor cells treated with BST/CaO$_2$ NSs exhibited distinct green fluorescence, indicating a significant increase in ROS production. These results suggest that the thermoelectric effect mediated by self-triggered BST/CaO$_2$ NSs has excellent potential for inducing ROS production. To confirm these findings, the intracellular fluorescence intensity of DCF under different treatments was analyzed using FCM to quantify the ROS generated by the thermoelectric effect mediated by BST/CaO$_2$ NSs (Supplementary Fig. 11). The ROS level in the BST/CaO$_2$ NS group was 73.62%, which was significantly higher than 0.48% in the control and 6.73% in the BST group (Fig. 5d), confirming that BST/CaO$_2$ NS-mediated thermoelectric catalysis can indeed accelerate intracellular ROS production.

There are two main mechanisms for the killing effect of ROS on tumor cells: one is to induce alteration of mitochondrial membrane potential leading to mitochondrial dysfunction; the other is to induce nuclear DNA damage in tumor cells. To investigate the mechanisms of the killing effect of ROS on tumor cells, JC-1 dye was used for mitochondrial membrane potential detection, and γ-H2AX staining was used for the DNA damage assay. As shown in Fig. 5e, treatment with BST and CaO$_2$ alone produced only slight mitochondrial membrane potential damage in CT26 cells, while substantial mitochondrial membrane potential damage was observed in the BST/CaO$_2$ NS treatment group. Similarly, significant DNA damage was specifically observed in BST/CaO$_2$ NS-treated CT26 cells as measured using a γ-H2AX probe (Fig. 5f). Furthermore, calcein AM/PI staining of dead cells (red) and live cells (green) revealed that BST/CaO$_2$ NS treatment induced a large amount of cell death in cancer cells (Fig. 5g). The extent of apoptosis was confirmed by FCM analysis (Fig. 5h and Supplementary Fig. 12), which was consistent with the results obtained by CLSM. These findings suggest that the self-triggered thermoelectric catalytic system BST/CaO$_2$ NSs can spontaneously generate thermoelectric effects in the TME, inducing a large amount of ROS production in tumor cells. The generated ROS caused a decrease in cellular mitochondrial membrane potential, leading to mitochondrial dysfunction and attacking the tumor cell nucleus, resulting in DNA damage and ultimately causing tumor cell death.

### In vivo imaging and biodistribution of BST/CaO$_2$ NSs

To evaluate the in vivo therapeutic performance of the BST/CaO$_2$ NS-based self-triggered thermoelectric system, CT26 xenograft tumor models were established in BALB/c mice. To investigate the biodistribution of the Cy5.5-labeled BST/CaO$_2$ NSs, they were intravenously injected into CT26 xenograft tumor models prior to evaluating their antitumor effect. The biodistribution of the BST/CaO$_2$ NSs was observed at 4, 12, and 24 h postinjection using in vivo imaging, and it was found that there was an effective and continuous accumulation of the nanoscale particles at the tumor site (Fig. 6a). This was further confirmed by semiquantitative analysis of BST/CaO$_2$ NSs in the major organs (including the heart, liver, spleen, lung, and kidney) and tumors 24 h after intravenous injection. As shown in Fig. 6b, a bright fluorescence signal was present in the dissected tumor, which was in agreement with the in vivo imaging results. Supplementary Fig. 13 shows the semiquantitative analysis of BST/CaO$_2$ NSs in the major organs and tumor 24 h after intravenous injection, which was in agreement with the in vivo imaging results, further demonstrating the EPR effect-induced accumulation of nanoscale BST/CaO$_2$ NSs at the tumor site. To more accurately characterize the distribution of the BST/CaO$_2$ NSs in vivo, photoacoustic (PA) imaging and computerized tomography (CT) were used to conduct real-time monitoring. Because of the excellent photothermal conversion performance of the BST NSs, they served as a PA indicator for in vivo photoacoustic imaging. Real-time

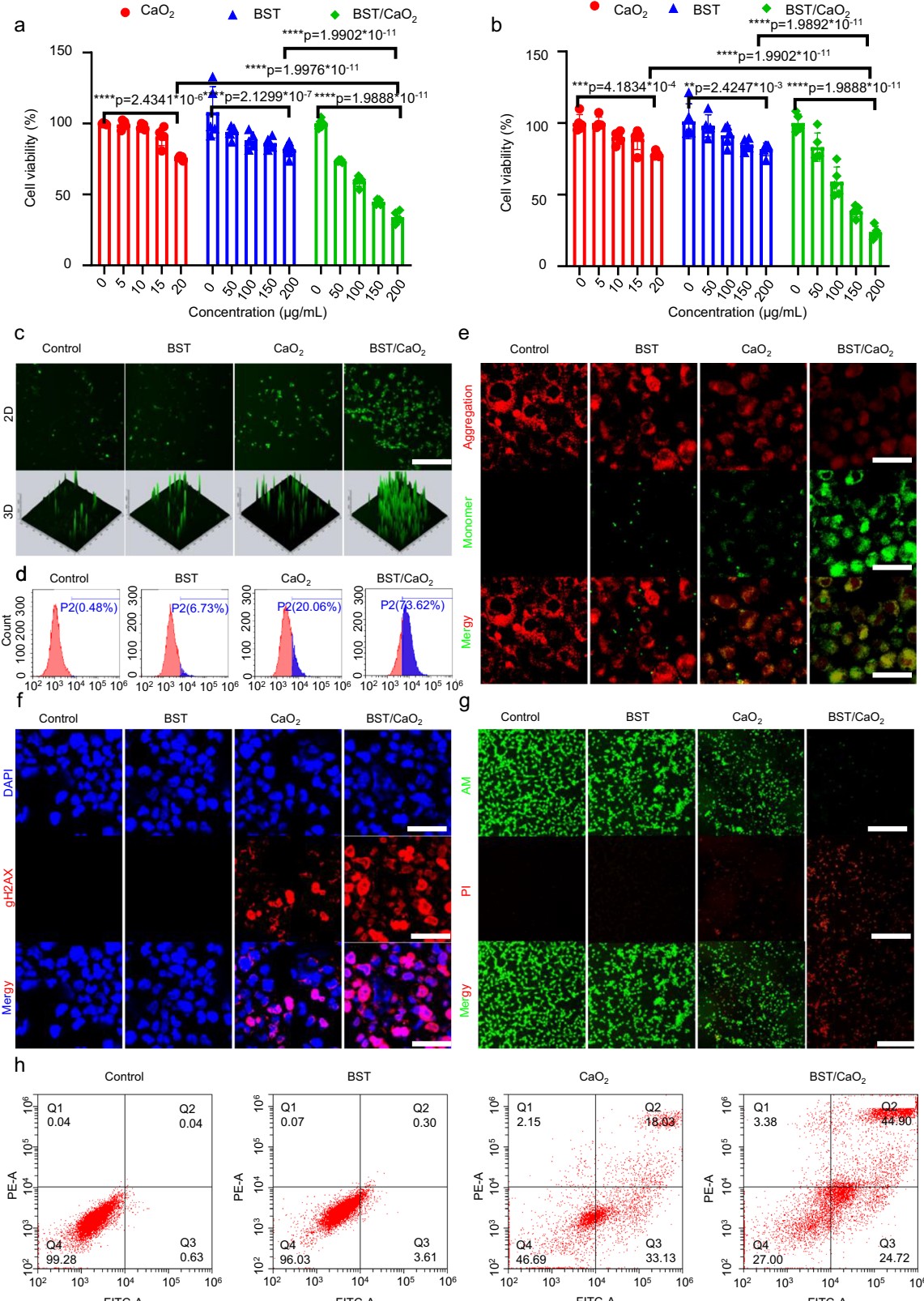

PA images of the tumor-bearing mice were recorded after intravenous injection with BST/CaO₂ NSs. The findings suggest that the BST/CaO₂ NS-based self-triggered thermoelectric system has great potential for use as a synergistic antitumor therapy in vivo due to its effective accumulation at the tumor site. As shown in Fig. 6c, BST/CaO₂ NSs accumulated in the tumor site well over time. Furthermore, it should

be noted that the BST/CaO₂ NSs also exhibit potential as CT imaging agents due to the high X-ray attenuation coefficient of Bi. In fact, as demonstrated in Fig. 6d, e, there is a positive correlation between the concentration of BST/CaO₂ NSs and the Hounsfield unit (HU) value, indicating their ability to serve as effective contrast agents for CT imaging. To evaluate their in vivo CT imaging potential, BST/CaO₂ NSs

**Fig. 5 | In vitro antitumor performance of the self-triggered thermoelectric system.** Cell viability of BST NSs, $CaO_2$ NPs and BST/$CaO_2$ NS-treated **a** CT26 cells and **b** TE1 cells by CCK8 assays. Data are presented as the mean ± s.d. ($n = 5$ biologically independent cells). Statistical differences were analyzed by Student's two-sided $t$-test. Representative fluorescence images and quantification of intracellular ROS by **c** CLSM and **d** FCM. Scale bar = 100 µm. Three times each experiment was repeated independently with similar results. **e** Representative confocal microscopy images of mitochondria-selective JC-1-stained CT26 cells after different treatments. Scale bar = 10 µm. Three times each experiment was repeated independently with similar results. **f** Representative confocal microscopy images of γ-H2AX-stained CT26 cells after different treatments. Scale bar = 10 µm. Three times each experiment was repeated independently with similar results. **g** Confocal imaging of CT26 cells stained with PI (red fluorescence) and Calcein-AM (green fluorescence) to distinguish dead cells and live cells after different treatments. Scale bar = 100 µm. Three times each experiment was repeated independently with similar results. **h** FCM images of CT26 cells stained with PI (red fluorescence) and Annexin V-FITC (green fluorescence) to measure cell apoptosis after treatment under different conditions.

were intravenously injected into CT26 tumor-bearing mice and analyzed using coronal CT imaging. The results, displayed in Fig. 6f, showed enhanced contrast within the tumor area, suggesting the potential for BST/$CaO_2$ NSs to serve as efficient CT imaging agents for cancer diagnosis. As shown by fluorescence and CT imaging in vivo, nanomaterials not only accumulate in tumors but also accumulate in the liver, kidney, spleen and lung. Because the liver, kidney and spleen are the main metabolic organs, nanomaterials are mainly excreted through the metabolism of the above three organs. The accumulation in the lungs is mainly due to the following reasons. The lung is a highly vascularized organ with a rich network of capillaries. The small size of the nanomaterials allows them to penetrate into lung tissue through the walls of blood vessels. The short distance between the alveoli and the pulmonary capillaries also promotes the deposition and aggregation of nanomaterials in the lungs[58,59]. In addition, the lung contains a large number of alveolar macrophages, which participate in the absorption and metabolism of foreign molecules and particles[60]. When the nanomaterials are injected intravenously, various serum proteins bind to the nanomaterials and are recognized and internalized by scavenger receptors on the surface of the macrophages, resulting in the aggregation of the nanomaterials in the lungs. However, it is eventually cleared out of the body by macrophages[61]. Moreover, to further investigate the biodistribution of BST/$CaO_2$ NSs in vivo, ICP/MS analysis was utilized, as depicted in Supplementary Fig. 14. The results indicated a significant accumulation of NSs within the major organs and tumors over a period of 30 days, highlighting their effectiveness in targeting tumors. Importantly, Supplementary Fig. 14 also illustrates that the accumulated BST/$CaO_2$ NSs within normal organs and tissues were gradually excreted by the body over time, indicating their biocompatibility and potential for clinical translation.

## In vivo antitumor effect of BST/$CaO_2$ NSs

Following the confirmation of high tumor accumulation, the in vivo therapeutic performance of the BST/$CaO_2$ NS-based self-triggered thermoelectric strategy was evaluated. As demonstrated in Fig. 7a, a schematic of BST/$CaO_2$ NS treatment in the tumor xenograft model is shown. Specifically, CT26 cells were subcutaneously injected into mice, and after 7 days, all mice were randomly divided into four groups: control, BST, $CaO_2$, and BST/$CaO_2$. Subsequently, mice were tail vein injected with PBS, BST NSs, $CaO_2$ NPs, or BST/$CaO_2$ NSs on days 1, 3, and 5 at a dosage of 5 mg/kg BST NSs and 0.5 mg/kg $CaO_2$ NPs. On day 12, the mice were sacrificed for further biological analysis. As illustrated in Fig. 7b, c, the results showed that the BST NSs alone did not exhibit significant tumor growth inhibition compared to the control (PBS) group due to the lack of the thermoelectric effect of BST NSs at a constant temperature. However, mice treated with $CaO_2$ NPs alone showed a certain extent of tumor growth inhibition. This was attributed to the decomposition of $CaO_2$ within the tumor TME, releasing $H_2O_2$ and $Ca^{2+}$ ions, which induced oxidative damage in cancer cells and may have caused ion interference therapy. Specifically, $H_2O_2$ could interfere with calcium ion channels, increasing the concentration of intracellular $Ca^{2+}$ in tumor cells, leading to cellular osmotic pressure and resulting in massive absorption and expansion. To further confirm the $Ca^{2+}$-mediated ion interference therapy, the concentration of $Ca^{2+}$ in tumor cells was measured. The results

demonstrated that each injection increased the $Ca^{2+}$ concentration in the subcutaneous graft tumor cells, which gradually returned to normal levels over time (Supplementary Fig. 15). Moreover, $Ca^{2+}$ was found to effectively promote the maturation and migration of DCs, enhancing antigen presentation ability and thus improving tumor immunotherapy. Compared to $CaO_2$ treatment alone, BST/$CaO_2$ treatment provided a superior therapeutic effect, indicating that the heat released from the decomposition of $CaO_2$ in the TME initiated the thermoelectric effect of BST, resulting in thermoelectric catalysis. Due to the synergistic effects of thermoelectric catalysis and immunoregulation exerted by BST/$CaO_2$ NSs, the BST/$CaO_2$ group exhibited the greatest inhibition of tumor growth and, correspondingly, the highest survival rate (Fig. 7f). Importantly, all treatments were virtually free of side effects, as no obvious weight loss was observed (Supplementary Fig. 16).

Additionally, an orthotopic colorectal cancer animal model was established by injecting CT26-luc cells ($2 \times 10^6$) into the colorectal wall of mice to evaluate the antitumor effect of the BST/$CaO_2$ NS-based self-triggered thermoelectric strategy. Once the bioluminescence intensity of the colon reached $1 \times 10^6$ photons (p) $s^{-1}$ $cm^{-2}$ $sr^{-1}$, the mice were randomly divided into four groups ($n = 5$) and received rectal perfusion of PBS, BST NSs, $CaO_2$ NPs, or BST/$CaO_2$ NSs three times a week. Three weeks later, the mice in each group received intraperitoneal injections of d-luciferin (150 mg kg$^{-1}$) and were anesthetized and imaged under the small animal imaging system to monitor tumor bioluminescence. As shown in Fig. 7d, the representative bioluminescence images demonstrated that BST NSs alone did not delay tumor growth compared to the control group. However, the tumor bioluminescence change images and curves in Fig. 7e revealed that $CaO_2$ NPs exhibited moderate in vivo anticancer effects during treatment. Encouragingly, BST/$CaO_2$ NSs showed a marked inhibitory effect on tumor growth compared to saline, demonstrating the high antitumor efficiency of the BST/$CaO_2$ NS-based self-triggered thermoelectric strategy in the orthotopic colorectal cancer animal model.

ROS explosion induces immunogenic cell death, releasing large amounts of tumor antigens[62–64]. The maturation and antigen presentation of DCs are crucial steps in determining the efficacy of immunotherapy[65–67]. To evaluate the maturation and migration of DCs to lymph nodes, FCM assays were performed on treated mouse tumors (Fig. 7g, h). The populations of mature DCs (CD80+CD86+) in both the $CaO_2$ NP and BST/$CaO_2$ NS groups were significantly increased by 3.8- and 4.1-fold compared to the control group, respectively (Fig. 7g and Supplementary Fig. 17). No significant changes in the populations of mature DCs were observed for the BST group, indicating that released $Ca^{2+}$ may be the most important factor triggering DC maturation. Subsequently, the migration of mature DCs to lymph nodes was analyzed, and the highest population of mature DCs was found in the BST/$CaO_2$ NSs group, which was 1.9-fold higher than that of the control group (Fig. 7h and Supplementary Fig. 18). Moreover, the levels of the inflammatory markers interferon-g (IFN-γ) and tumor necrosis factor-α (TNF-α) in serum were significantly increased by 1.9- and 1.6-fold in the BST/$CaO_2$ NSs group compared to the control and BST groups, respectively (Supplementary Figs. 19 and 20). To confirm the role of $Ca^{2+}$ in promoting the maturation of DC cells, we measured the concentration of $Ca^{2+}$ in lymphocytes at lymph nodes. As depicted in

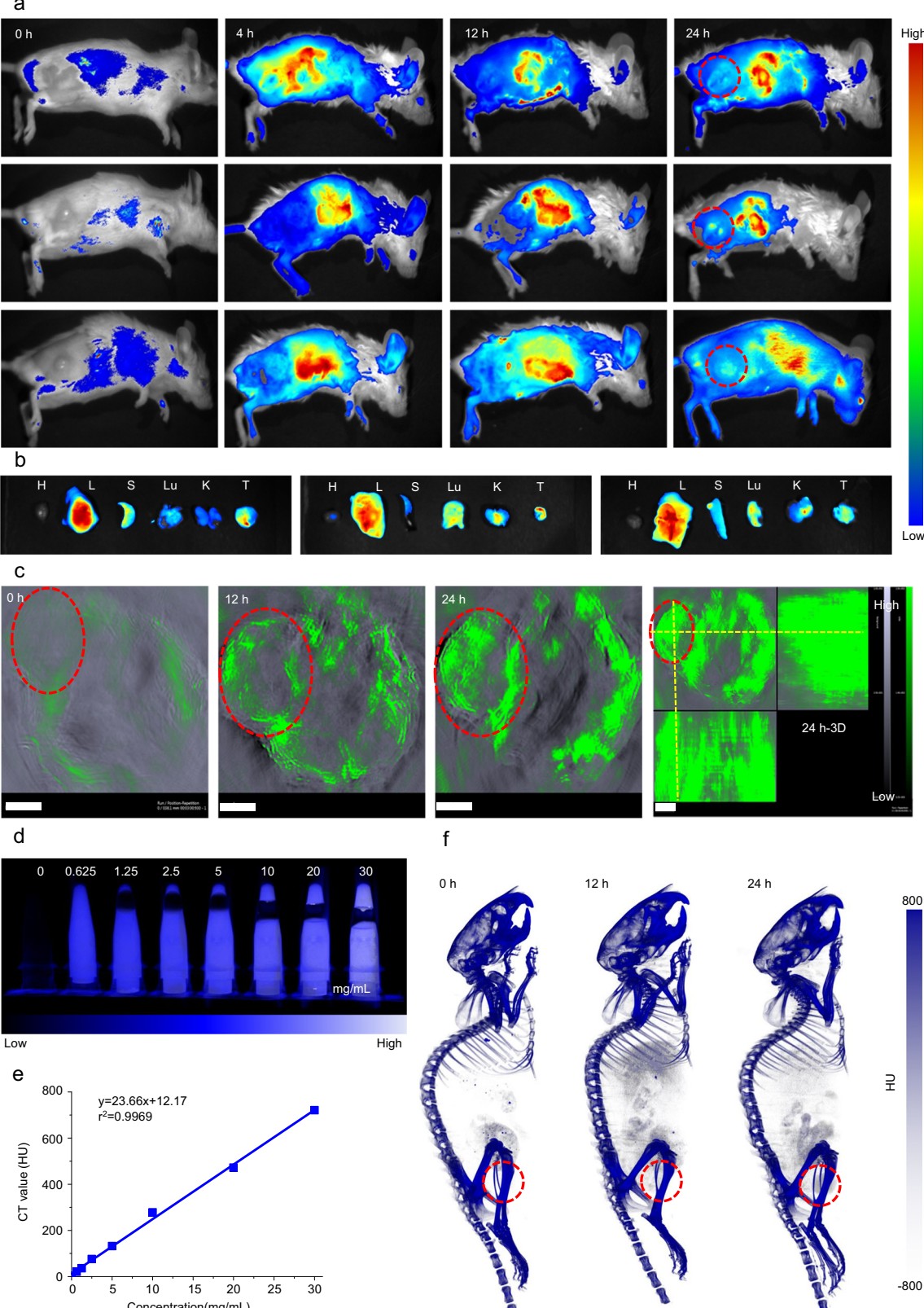

**Fig. 6 | In vivo imaging and biodistribution of the BST/CaO$_2$ heterojunction. a** In vivo fluorescence images of tumor-bearing mice at different time points after intravenous injection with Cy5.5-labeled BST/CaO$_2$ heterojunction and **b** ex vivo fluorescence images of tumor and major organs at 24 h after injection. **c** In vivo photoacoustic images of tumor-bearing mice after intravenous injection with BST/ CaO$_2$ heterojunction. Scale bar = 3 mm. Three times each experiment was repeated independently with similar results. **d** CT images of BST/CaO$_2$ heterojunctions with different concentrations. **e** The CT values (HU) of the BST/CaO$_2$ heterojunction. **f** In vivo CT images of tumor-bearing mice after intravenous injection with the BST/ CaO$_2$ heterojunction. The red circle indicates a tumor. Three times each experiment was repeated independently with similar results.

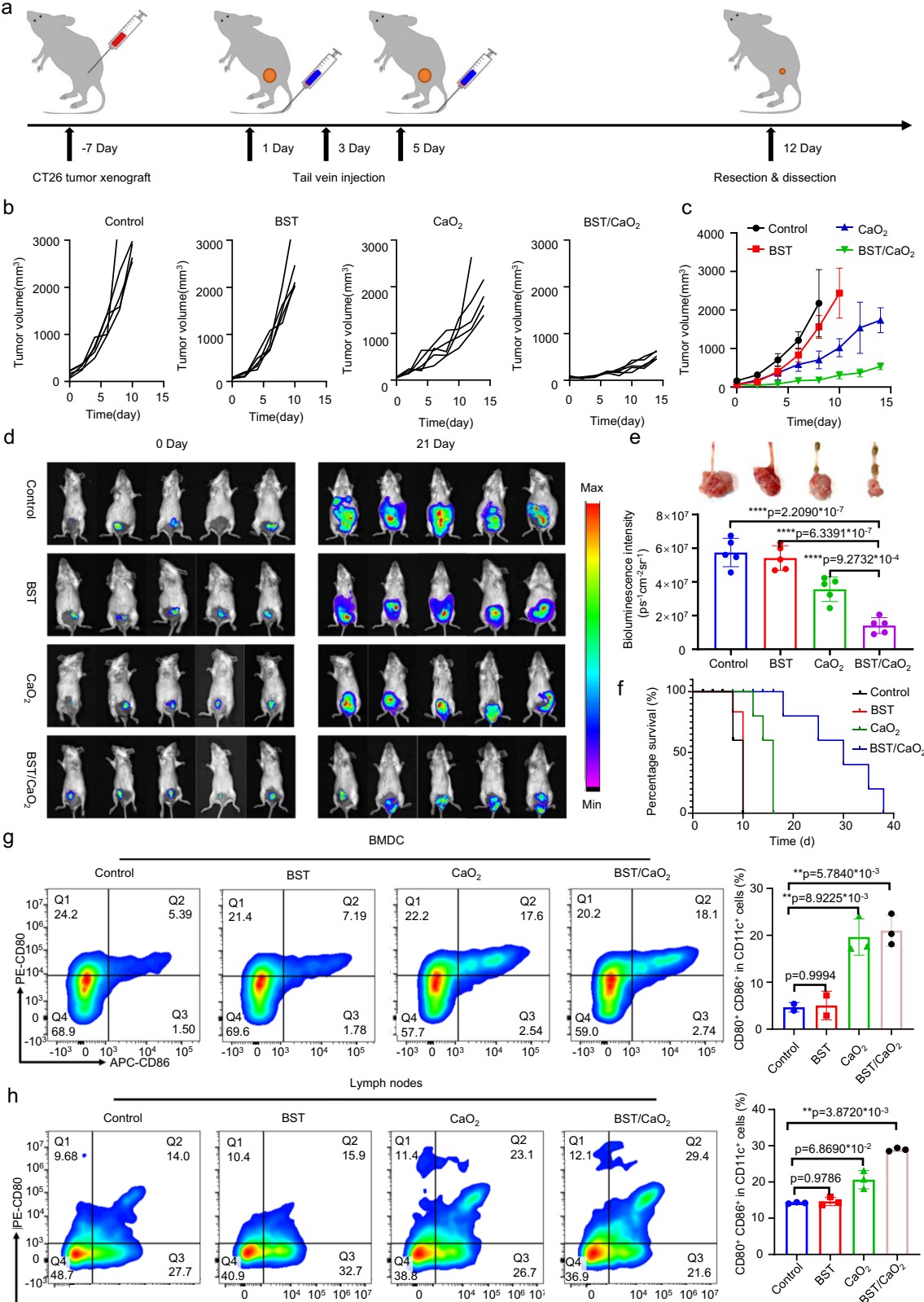

**Fig. 7 | In vivo antitumor performance of the self-triggered thermoelectric system. a** Experimental illustration of in vivo antitumor therapy. **b, c** Antitumor performance of different treatments, including control, BST, $CaO_2$ and BST/$CaO_2$, on a subcutaneous xenograft colorectal cancer animal model. Data are presented as the mean ± s.d. ($n$ = 5 biologically independent mice). Statistical differences were analyzed by Student's two-sided $t$-test. **d, e** Antitumor performance of different treatments, including control, BST, $CaO_2$ and BST/$CaO_2$, on an orthotopic colorectal cancer animal model. Data are presented as the mean ± s.d. ($n$ = 5 biologically independent mice). Statistical differences were analyzed by Student's two-sided $t$-test. **f** The survival curves of tumor-bearing mice under different treatments. **g, h** Flow cytometry analysis of the percentage of DC maturation and migration to lymph nodes ($CD11c^+$ $CD80^+$ $CD86^+$) under different treatments. Data are presented as the mean ± s.d. ($n$ = 3 biologically independent mice). Statistical differences were analyzed by Student's two-sided $t$-test.

Supplementary Fig. 15, the concentration of $Ca^{2+}$ in lymphocytes was positively correlated with the number of injections, indicating that an increase in $Ca^{2+}$ concentration could be detected in lymphocytes 12 h after each injection. However, over time, the concentration of $Ca^{2+}$ in the lymphocytes gradually returned to normal levels.

To further investigate the underlying mechanism, tumor sections treated with BST/CaO$_2$ NSs were analyzed using immunofluorescence (IF) staining of γ-H2AX and cleaved caspase-3 (C-CAS3) as markers for DNA double-strand breakage and cell apoptosis, respectively. As shown in Fig. 8a and Supplementary Fig. 21, high levels of irreparable DNA damage and cell apoptosis were observed in tumor sections treated with BST/CaO$_2$ NSs due to the self-triggered thermoelectric catalysis and $Ca^{2+}$-induced immunoregulation of the particles. Additionally, the self-triggered thermoelectric and immunotherapy of BST/CaO$_2$ NSs was confirmed through TUNEL staining, which revealed a larger area of apoptosis in cancer cells after treatment with BST/CaO$_2$ NSs (Fig. 8c). These findings demonstrate the efficient and synergistic effects of the self-triggered thermoelectric and immunotherapy of BST/CaO$_2$ NSs. Furthermore, to obtain a more accurate quantitative measurement of apoptosis and DNA damage in organs and tumors in each treatment group, we used FCM to conduct a detailed analysis. As depicted in Fig. 9 and Supplementary Figs. 22 and 23, no significant DNA damage or apoptosis was observed in normal organs (heart, liver, spleen, lung, kidney) across all treatment groups. However, BST/CaO$_2$ NSs showed significant DNA damage and apoptosis in tumor tissues, confirming the tumor specificity and biosafety of the self-triggered thermoelectric strategy based on BST/CaO$_2$ NSs.

### In vivo biosafety evaluation of BST/CaO$_2$ NSs

The in vivo toxicity of nanomedicine is a crucial factor for translation from bench to practical applications. Therefore, we meticulously investigated the toxicity of BST/CaO$_2$ NSs through histology examination, hematology assay, and immune analysis. Although three doses of BST/CaO$_2$ NSs were injected into each mouse, the concentrations of both BST NSs and CaO$_2$ NPs were very low, at 0.1 and 0.01 mg per mouse, respectively. The concentrations of $Ca^{2+}$ in blood exhibited no significant fluctuation due to the slow hydrolysis of CaO$_2$ NPs under normal conditions (Supplementary Fig. 15). As shown in Fig. 6 and Supplementary Fig. 14, BST/CaO$_2$ NSs were distributed to a certain extent in the liver, spleen, lung, and other important organs, with their distribution decreasing over time. We hypothesize that the BST/CaO$_2$ NSs will be partially excreted through renal urination and intestinal defecation, which was confirmed by ICP/MS analysis of urinary and fecal excretion (Supplementary Fig. 24). This alleviated the long-term retention of BST/CaO$_2$ NSs in the body to some extent. The IF staining of major organs, such as the heart, liver, spleen, lung, and kidney, of BST/CaO$_2$ NS-treated mice with γ-H2AX and C-CAS3 as markers of DNA double-strand breaks and cell apoptosis indicated no significant apoptosis or DNA damage in these normal organs (Fig. 8b, d). Moreover, TUNEL staining of normal organs of mice treated with BST/CaO$_2$ NSs confirmed their biosafety as an antitumor strategy. Furthermore, FCM analysis revealed no detectable DNA damage and apoptosis in the heart, liver, spleen, lung, and kidney for each treatment group compared to obvious DNA damage and apoptosis in the tumors (Fig. 9). Overall, these results suggest the biosafety of BST/CaO$_2$ NS-based antitumor strategies.

Hematoxylin and eosin (H&E) staining of major organs after treatment with BST/CaO$_2$ NSs was carried out to further confirm the biocompatibility and specific targeted antitumor mechanism of BST/CaO$_2$ NS-based therapy. As depicted in Fig. 10a, although intravenously injected BST/CaO$_2$ NSs partially accumulated in normal organs (mainly in the liver, spleen, and lung), they caused almost no damage. Moreover, real-time quantitative PCR (RT–qPCR) was applied to detect the damage and inflammatory response of each major organ exposed to BST/CaO$_2$ NSs. The results presented in Fig. 10b confirmed the good

biocompatibility and biosafety of BST/CaO$_2$ NS-based cancer therapy. We also performed hematological detection to investigate the systematic biosafety properties of BST/CaO$_2$ NSs. The mean corpuscular hemoglobin concentration (MCHC), red blood cells (RBCs), hematocrit (HCT), white blood cells (WBCs), platelets (PLTs), mean corpuscular volume (MCV), hemoglobin (HGB), and mean corpuscular hemoglobin (MCH) were measured (Fig. 10c). There was no statistically significant difference observed in the BST/CaO$_2$ NS-treated groups 7 and 14 days after i.v. injection compared to the control group. Furthermore, blood biochemical parameters, including γ-glutamyl transpeptidase (γ-GT), total protein (TP), C-reactive protein (CRP), creatine kinase (CK), lactate dehydrogenase (LDH), creatinine (Cr), blood urea nitrogen (BUN), alanine aspartate aminotransferase (AST), amylase (AMY), aminotransferase (ALT), and albumin (ALB), between the control mice and the mice injected with BST/CaO$_2$ NSs for 1, 7, and 14 days were tested. As presented in Fig. 10d, there were nearly no observable differences between the BST/CaO$_2$ NSs and control groups. Therefore, all of the aforementioned results demonstrate that our prepared BST/CaO$_2$ NSs should be considered a relatively biosafe and biocompatible nanomedicine.

## Discussion

Catalytic therapies are a promising approach to cancer treatment, utilizing nanocatalysts that are nontoxic or low toxic to convert intracellular O$_2$ or H$_2$O into ROS, such as $\cdot O_2^-$, $\cdot OH$, and H$_2$O$_2$. These ROS induce effective tumor-specific oxidative damage and apoptosis without causing significant toxicity to normal organs or tissues[1,11,12]. However, most catalytic therapies rely on exogenous excitation, which means they require specific external stimuli such as light or ultrasound to trigger catalytic reactions. Unfortunately, exogenous excitation catalytic therapy faces several challenges in clinical application. First, the penetration of light is limited, and high-intensity ultrasound may cause collateral mechanical damage. Second, the catalytic efficiency of these therapies is often low due to the fast recombination of excited holes and electrons. Last, the use of additional excitation equipment can add operational complexity and inconvenience to the treatment process[11,51,52].

Recently, a type of catalytic therapy has been developed for cancer treatment that combines thermoelectric effects and redox reactions[37,38]. This approach is different from photocatalysis and piezocatalysis because it uses temperature fluctuation to generate pyro-generated negative and positive charges, which can trigger chemical oxidation–reduction reactions. Thermoelectric catalysis involves the creation of a self-built-in electric field inside a thermoelectric catalyst, which retards electron-hole recombination and allows for greater catalytic activity and higher ROS generation[43]. However, current thermoelectric catalysis is triggered by laser irradiation through photothermal conversion, which limits its penetration into biological tissue and reduces its catalytic efficiency. To address this issue, there is a need to develop self-triggered thermoelectric catalytic materials or systems that maintain the advantages of thermoelectric catalysis while avoiding its limitations. Such developments hold great promise for clinical applications.

In this study, we have presented a self-triggered thermoelectric nanoheterojunction for enhanced tumor catalytic therapy and coupling with immunotherapy. We synthesized a conventional and efficient thermoelectric biomaterial, BST NSs, and selected it as the thermoelectric catalyst. The innovation lies in the in situ coating of CaO$_2$ NPs, which not only acted as a trigger in response to the TME but also activated the immune system and imported immunotherapy. We explain that the CaO$_2$ NPs were hydrolyzed rapidly into $Ca^{2+}$ and H$_2$O$_2$ in the acidic TME, generating a large amount of heat. The thermoelectric effect of BST NSs was activated by heat, producing negative and positive charges for chemical oxidation–reduction reactions and ROS generation. The self-built-in electric field inside BST NSs guided the separation of electron-hole pairs and retarded electron-hole

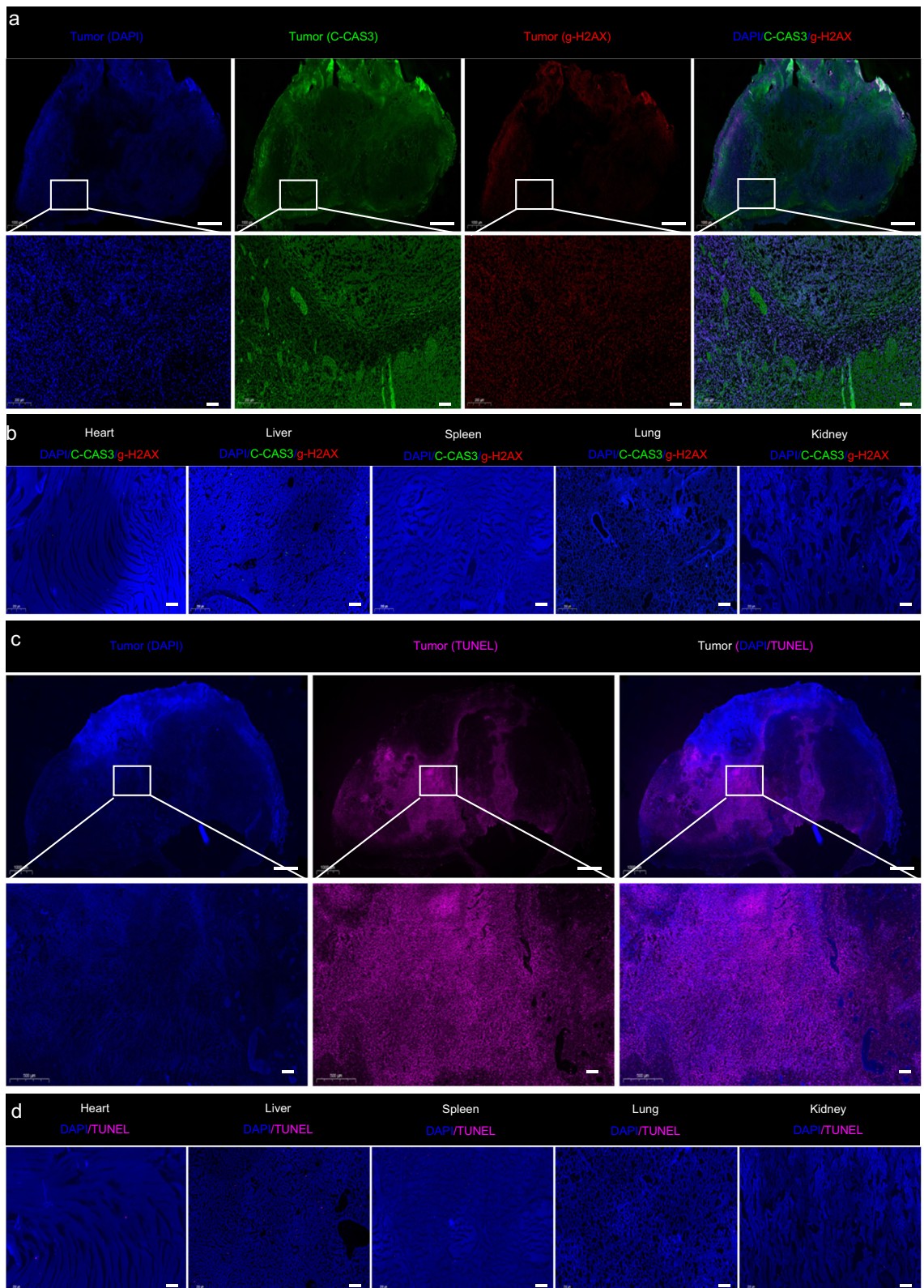

**Fig. 8 | In vivo immunofluorescence staining and tissue damage analysis.**
**a**, **b** Immunofluorescence images of the tumors and major organs (heart, liver, spleen, lung, and kidney) obtained from mice injected with BST/CaO$_2$ NSs. The nucleus was stained with DAPI (blue), the damaged DNA was stained with γ-H2AX foci (red), and the apoptotic cells were stained using the apoptosis marker C-CAS3 (green). Scale bars: 1000 μm for the first line and 100 μm for the second and third lines. **c**, **d** TUNEL staining of the tumors and major organs (heart, liver, spleen, lung, and kidney) obtained from mice injected with BST/CaO$_2$ NSs. Scale bars: 1000 μm for the first line and 100 μm for the second and third lines. Three times each experiment was repeated independently with similar results.

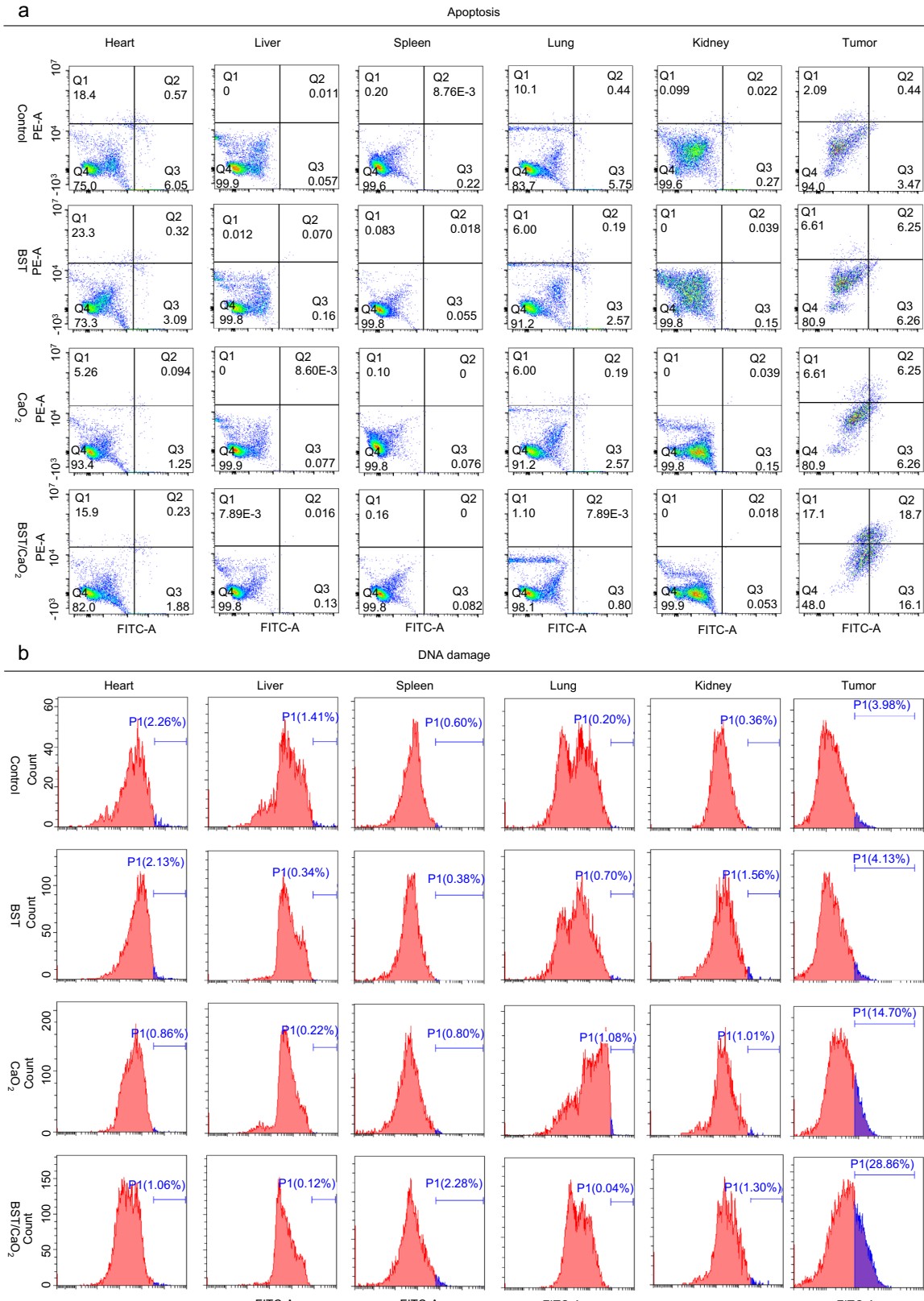

**Fig. 9 | Quantitative analysis of apoptosis and DNA damage in major organs (heart, liver, spleen, lung, and kidney) and tumors under different treatments by FCM on the tenth day of treatment. a** Apoptosis in the heart, liver, spleen, lung, kidney, and tumor under different treatments. **b** DNA damage in the heart, liver, spleen, lung, kidney, and tumor under different treatments.

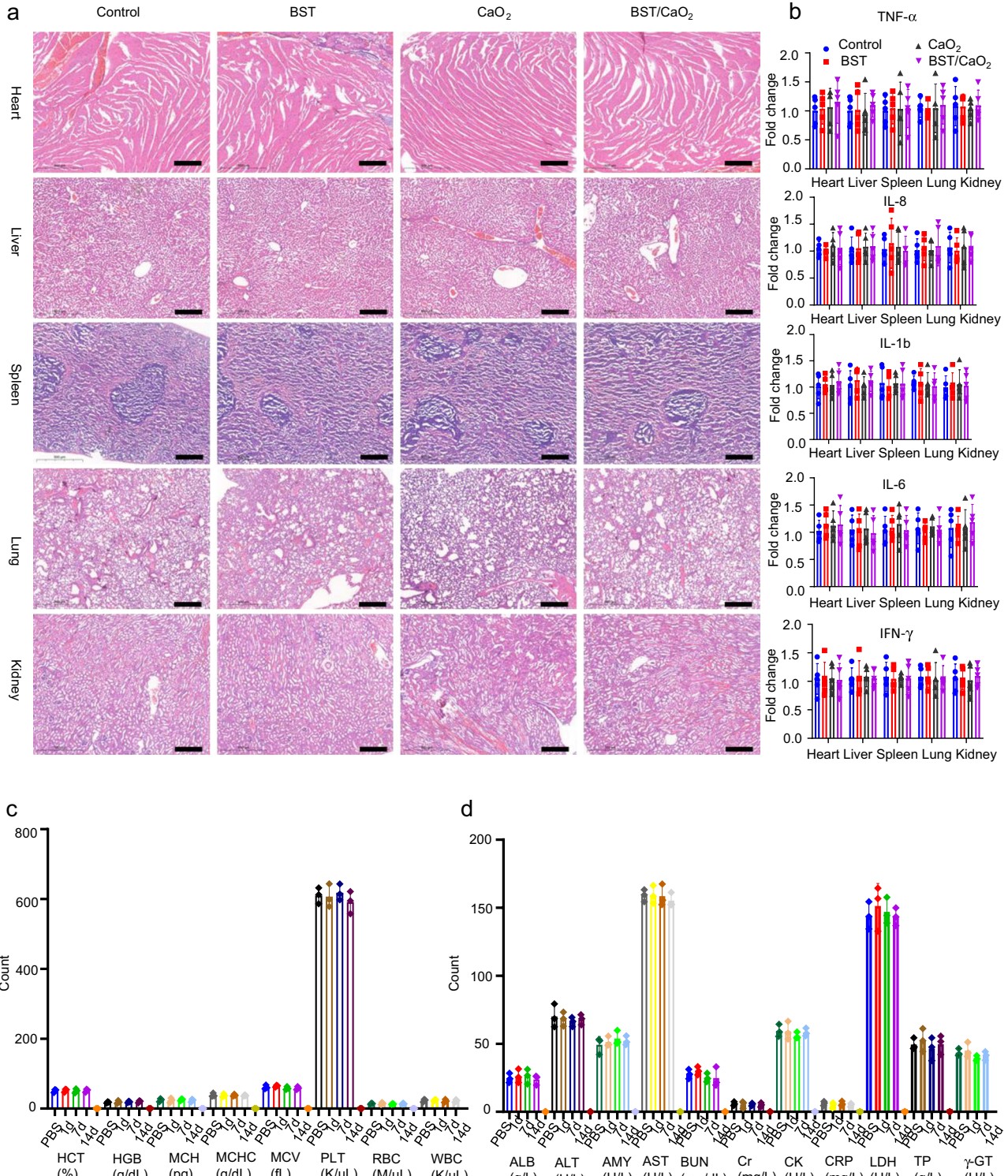

**Fig. 10 | Biosafety assessment of BST/CaO₂ NS-based self-triggered thermo-electric and immunological therapy. a** Representative H&E-stained images of the heart, liver, lung, kidney, and spleen under different treatments. Scale bars = 300 μm. Three times each experiment was repeated independently with similar results. **b** Analysis of inflammatory factors in the heart, liver, lung, kidney, and spleen under different treatments. Data are presented as the mean ± s.d. (*n* = 5 biologically independent mice). Statistical differences were analyzed by Student's two-sided *t*-test. **c** Blood hematology analysis of Balb/c mice under different treatments. Data are presented as the mean ± s.d. (*n* = 5 biologically independent mice). Statistical differences were analyzed by Student's two-sided *t*-test. **d** Blood biochemical analysis of Balb/c mice under different treatments. Data are presented as the mean ± s.d. (*n* = 5 biologically independent mice). Statistical differences were analyzed by Student's two-sided *t*-test.

recombination. $H_2O_2$ provided substrate ($O_2$) supplementation for thermoelectric catalysis and ROS production and regulated $Ca^{2+}$ channels while delaying $Ca^{2+}$ efflux. The main hydrolysate $Ca^{2+}$ could mediate ion interference therapy, breaking intracellular ionic homeostasis and increasing the osmotic pressure of tumor cells. Additionally, it could promote DC maturation and tumor antigen presentation, thus activating an immune response and mediating effective immunotherapy. Moreover, the $CaO_2$ NP coating hydrolyzed slowly in normal cells to generate $Ca^{2+}$ and $O_2$, where the slowly released $Ca^{2+}$ was expelled from the cells via calcium channel proteins. Without the trigger of temperature fluctuation, the BST NSs possessed excellent biosafety and biocompatibility in normal organs and tissues. Overall, this study provides an intelligent strategy for the synthesis of a tumor-specific self-triggered thermoelectric catalyst and provides insights into an advanced strategy to enhance the application scope and efficiency of catalytic therapy. Tumor-specific self-triggered thermoelectric catalysis based on BST/$CaO_2$ heterojunction combined catalytic therapy, ion interference therapy, and immunotherapy exhibited excellent antitumor and biosafety properties both in vitro and in vivo.

Although the self-triggered synergistic thermoelectric, ionic interference, and immunotherapy demonstrated in this study show promising advantages and potential applications, further research is needed before clinical use. For instance, a more detailed and comprehensive analysis of material metabolic pathways and toxicological implications should be conducted. Additionally, the thermoelectric materials based on BST NSs used in this study degrade slowly in vivo, leading to accumulation in vital organs. Although no adverse reactions were detected during the short-term study, it is challenging to predict long-term residual toxicity in vivo. Therefore, developing safe and efficient degradable thermoelectric materials coupled with tumor-specific switches, such as hypoxia-responsive, low pH-responsive, and high ROS-responsive materials, would be a crucial strategy to promote the clinical transformation of self-triggered thermoelectric dynamic therapy.

## Methods

### Ethical statement
This research complies with all relevant ethical regulations. All animal studies were conducted in accordance with the National Institute Guide for the Care and Use of Laboratory Animals. The experimental protocols were approved by the Animal Ethics Committee of the Tianjin University Laboratory Animal Center (Tianjin, China) (Approval No. TJUE-2022-210).

### Animal study
BALB/c mice (female, 6 weeks, 14–16 g) were used in this study. Animals were raised in specific pathogen-free animal experimental center and allowed free access to food and water. All experimental/control animals were co-housed in a habitant under standard conditions (23–26 °C, 40–60% humidity, 12 h light–dark cycle, and 3–4 mice or rats/cage). At the endpoint of the study, animal euthanasia was performed by $CO_2$ inhalation followed by cervical dislocation.

### Materials
Bismuth (III) nitrate pentahydrate [$Bi(NO_3)_3 \cdot 5H_2O$], antimony chloride ($SbCl_3$), sodium tellurite ($Na_2TeO_3$), sodium hydroxide (NaOH), polyvinyl pyrrolidone (PVP, MW: 10000), polyacrylic acid (PAA, MW: 3000) and 1,3-diphenylisobenzofuran (DPBF) were purchased from Sigma–Aldrich. The DNA Damage Assay Kit (γ-H2AX Immunofluorescence), 2,7-dichlorodihy-drofluorescein diacetate (DCFH-DA), and Calcein/PI Cell Viability/Cytotoxicity Assay Kit were purchased from Beyotime. The JC-1 Mitochondrial Membrane Potential Assay Kit was obtained from AbMole. DMEM, fetal bovine serum (FBS), trypsin-EDTA, and phosphate-buffered saline (PBS, pH 7.4 and pH 5.5) were supplied by

Gibco Life Technologies. Antibodies, including anti-CD11c, anti-CD86, and anti-CD80, were supplied by BD Pharmingen.

### Preparation of BST NSs
To prepare the reaction mixture, 1.5 mmol of $SbCl_3$, 0.5 mmol of $Bi(NO_3)_3 \cdot 5H_2O$, and 3 mmol of $Na_2TeO_3$ were added to 40 mL of ethylene glycol. Then, 2 mL of a 5 mol/L NaOH solution and 200 mg of polyvinyl pyrrolidone (PVP) were also added to the mixture. After stirring vigorously for 30 min, the homogeneous solution was transferred to a 100 mL Teflon reaction vessel and sealed. The reaction vessel was then heated in a microwave oven at 230 °C for 8 h. Once the reaction had finished, the resulting solutions were washed, centrifuged (1760×$g$) three times, and dried at 60 °C for 12 h in a vacuum oven.

### Preparation of $CaO_2$ NPs
$CaCl_2$ (0.1 g) and PVP (0.35 g) were weighed into a round flask and dissolved in 15 mL of ethanol using an ultrasound device. While stirring, 1 mL of ammonia and 0.2 mL of $H_2O_2$ solution (30 v/v %) were slowly added to the mixture to obtain a light blue milky white solution. The resulting product was collected by centrifugation at 15846×$g$, washed three times with ethanol, and finally redispersed in deionized water.

### Preparation of ligand-free BST NSs
Five milligrams of the BST NSs were added to 5 mL of deionized water. Then, 2 mL of a 0.5 M HCl solution was added to the mixture, and it was sonicated for 10 min. After sonication, the mixture was centrifuged at 15846×$g$ for 5 min. The resulting precipitates were then collected and dispersed in deionized water. This process was repeated three times using centrifugation to wash the nanoparticles. Finally, the collected nanoparticles were redispersed in deionized water.

### Preparation of BST/$CaO_2$ NSs
To prepare the BST/$CaO_2$ nanoparticles, 150 mg of $CaCl_2$ and 400 mg of PAA were dissolved in 10 mL of deionized water. Next, 5 mL of ligand-free BST nanoparticles at a concentration of 1 mg/mL were added to the solution with stirring. The resulting mixture was then sonicated for 30 min. After sonication, 200 μL of $H_2O_2$ was added to the solution, and it was stirred for an additional 30 min. The BST/$CaO_2$ nanoparticles were then collected by centrifugation for 5 min at 15846×$g$ and washed three times with deionized water. Finally, the BST/$CaO_2$ NSs were redispersed in deionized water and stored for further use.

### Characterization of BST NSs, $CaO_2$ NPs and BST/$CaO_2$ NSs
The morphologies of BST NSs, $CaO_2$ NPs and BST/$CaO_2$ NSs were investigated by transmission electron microscopy (TEM, TecnaiG2F20, Holland) and scanning electron microscopy (SEM, FEI, Czech Republic). followed by elemental analysis was performed (EDS, FEI, Czech Republic). The zeta potentials and hydrodynamic diameters were detected using a laser granularity analyzer (Litesizer 500, Austria). X-ray photoelectron spectroscopy (XPS, ECALAB250Xi, United States) was used to test the composition of the NSs. X-ray diffraction (XRD, D8 Advance, Germany) was applied to analyze the chemical structure of the materials.

### Computational model and details
To investigate the catalytic mechanism of BST, first-principles calculations based on density functional theory (DFT) were performed using the VASP package (VASP 5.4). The Perdew-Burke-Ernzerhof (PBE) formalism of the generalized gradient approximation (GGA) was considered here. The DFT-D2 method of Grimme correction was chosen to describe the van der Waals force. An R-3 m $Bi_2Te_3$ supercell model containing 20 atoms with a 15 Å vacuum layer was used to construct three possible BST models, and the model with the lowest total energy was adopted for further study. The Brillouin zone integration was

sampled by $6 \times 6 \times 1$ Monkhorst-Pack $k$-point meshes. The cutoff energy was set to 500 eV, and the convergence threshold was set to $1 \times 10^{-5}$ Ha in energy and $1 \times 10^{-2}$ Ha/Å in force. The initial binding site for $O_2$ was searched by Monte Carlo (MC) annealing simulations, which allow a rotatable molecule to randomly translate on the surface of the substrate until the local energy minima were reached.

Bader's charge analysis was performed to estimate the charge transfer ($\Delta Q$) between the substrate and adsorbate. $\Delta Q$ can be defined as

$$\Delta Q = Q - Q_0 \tag{1}$$

where $Q_0$ and $Q_1$ are the number of electrons occupied by the adsorbate before and after adsorption, respectively. Based on the definition, a positive magnitude of $\Delta Q$ infers electron flow from the substrate to adsorbate.

## Temperature changes at different pH values
To determine the trigger effect of $CaO_2$ NPs, the temperature change of BST NSs, $CaO_2$ NPs and BST/$CaO_2$ NSs solutions was recorded. In detail, BST NSs, $CaO_2$ NPs and BST/$CaO_2$ NSs were prepared and added into PBS solutions (final concentration 0.1 mg/mL) with different pH (pH 7.4 and pH 5.5), respectively. The temperature of the solution was detected and recorded by an infrared thermometer.

## $\cdot O_2^-$ production at different pH values
To determine the $\cdot O_2^-$ generation capability of BST/$CaO_2$ NSs, 1,3-diphenylisobenzofuran (DPBF) was used. The BST/$CaO_2$ NSs were dispersed in pH 5.5 and pH 7.4 PBS solution at a final concentration of 0.1 mg/mL. Then, 150 μL of DPBF (2 mg/mL) was added to each solution, and the mixture was stirred. The absorbance of DPBF at different time points was recorded using a UV–vis-NIR spectrophotometer.

## Cell line sources
Human normal liver cells (HL-7702, catalog number:77402), human embryonic kidney cells (HEK 293, catalog number: CRL-1573), mouse colorectal cancer cells (CT26, catalog number: CRL-2638) and human esophageal carcinoma cells (TE1, catalog number: PDM-225) were obtained from the American Type Culture Collections (ATCC).

## Antibodies
**Primary antibodies.** Phospho-Histone H2AX (Ser139) (D7T2V), Cell Signaling (Product # 80312), Dilution 1:200; Cleaved Caspase-3 (Asp175) (5A1E), Cell Signaling (Product # 9664), Dilution 1:250; Rat anti mouse CD11c, Miltenyi Biotec (Product 130-128-247), clone number: REA618, Dilution 1:200; Rat anti mouse CD86, Miltenyi Biotec (Product 130-102-558), clone number: PO3.3, Dilution 1:100; Rat anti mouse CD80, Miltenyi Biotec (Product 130-117-683), clone number: 2D10, Dilution 1:100.

**Secondary antibodies.** Goat Anti-Rabbit IgG (H+L) Highly Cross-Adsorbed Secondary Antibody, Alexa Fluor 488, ThermoFisher (Catalog # A-11034), Dilution 1:1000; Goat Anti-Mouse IgG (H+L) Highly Cross-Adsorbed Secondary Antibody, Alexa Fluor 647, ThermoFisher (Catalog # A-21236), Dilution 1:1000.

## Validation
**Primary antibodies.** Phospho-Histone H2AX (Ser139) (D7T2V), Cell Signaling (Product # 80312), Dilution 1:200; IHC-Leica® Bond™ 1:200–1:800; Immunohistochemistry (Paraffin) 1:200–1:800; Immunofluorescence (Immunocytochemistry) 1:100–1:400; Flow Cytometry; Species Reactivity: Human, Mouse, Rat, Monkey. Cleaved Caspase-3 (Asp175) (5A1E), Cell Signaling (Product # 9664), Dilution 1:250, validate for Western Blotting 1:1000; Immunoprecipitation 1:50; Immunohistochemistry (Paraffin) 1:2000; Immunofluorescence (Immunocytochemistry) 1:400–1:1600; Flow Cytometry; Species

Reactivity: Human, Mouse, Rat, Monkey. Rat anti-mouse CD11c, Miltenyi Biotec (Product 130-128-247), clone number: REA618, Dilution 1:200; Flow Cytometry 1:200; Species Reactivity: Human. Rat anti-mouse CD86, Miltenyi Biotec (Product 130-102-558), clone number: PO3.3, Dilution 1:100; Flow Cytometry 1:100; Species Reactivity: Mouse. Rat anti-mouse CD86, Miltenyi Biotec (Product 130-102-558), clone number: PO3.3, Dilution 1:100; Flow Cytometry 1:100; Species Reactivity: Human.

**Secondary antibodies.** Goat Anti-Rabbit IgG (H+L) Highly Cross-Adsorbed Secondary Antibody, Alexa Fluor 488, ThermoFisher (Catalog # A-11034), Dilution 1:1000; Immunohistochemistry (IHC), Immunohistochemistry (Paraffin) (IHC (P)), Immunohistochemistry (Frozen) (IHC (F)), Immunohistochemistry–Free Floating (IHC (Free)), Flow Cytometry (Flow), Immunocytochemistry (ICC/IF), Miscellaneous PubMed (Misc).

## CCK8 assays
HL-7702, HEK293, CT26, and TE1 cells were seeded on 96-well microplates ($10^4$ cells for each well) for 24 h (37 °C, 5% $CO_2$). Cells were then incubated with different concentrations (ranging from 50 to 200 μg/mL) of BST NSs, $CaO_2$ NPs, and BST/$CaO_2$ NSs for 24 h, and a CCK8 assay was conducted according to the manufacturer's protocol to measure the viability of cells.

## Intracellular $Ca^{2+}$ detection
Intracellular $Ca^{2+}$ production can be characterized by Fluo-4 AM. In detail, CT26 cells were seeded on 96-well plates ($10^4$ cells for each well) for 24 h (37 °C, 5% $CO_2$). Afterward, the cells were treated with different nanoparticles, including BST NSs, $CaO_2$ NPs, and BST/$CaO_2$ NSs, at a final concentration of 0.1 mg/mL for NSs and 0.01 mg/mL for NPs. Following a 12-h treatment period, the CT26 cells were washed three times with PBS and stained with Fluo-4 AM probe for 20 min. Finally, intracellular $Ca^{2+}$ production was measured using CLSM and FCM analysis according to the manufacturer's protocol.

## Intracellular ROS detection
The ROS probe DCFH-DA was used to investigate the intracellular ROS level via CLSM and FCM analysis. Briefly, CT26 cells were seeded into 6-well microplates ($10^6$ cells for each well) for 24 h (37 °C, 5% $CO_2$). Twenty-four hours following incubation with BST NSs, $CaO_2$ NPs and BST/$CaO_2$ NSs, CT26 cells were washed and incubated with 0.2 μM DCFH-DA solution (S0033; Beyotime) for 1 h at 37 °C. After rinsing with dilution buffer, the intracellular ROS level was measured using CLSM and FCM analysis according to the manufacturer's protocol. Analysis of FCM data was performed with Flowjo v7.6 software.

## Intracellular DNA damage detection
To determine early DNA damage, phosphorylated H2AX was analyzed using the Phospho-Histone H2AX (Ser139) antibody. CT26 cells were seeded onto 6-well plates ($10^6$ cells for each well) and allowed to grow for 24 h at 37 °C with 5% $CO_2$. Afterward, the cells were treated with different nanoparticles, including BST NSs, $CaO_2$ NPs, and BST/$CaO_2$ NSs, at a final concentration of 0.1 mg/mL. Following a 12-h treatment period, the CT26 cells were washed three times with PBS and stained with Phospho-Histone H2AX (Ser139) antibody. After an additional 30-min incubation, the stained CT26 cells were washed three times with PBS, and cellular phosphorylated H2AX (γ-H2AX) levels under different treatments were determined using CLSM.

## Intracellular mitochondrial membrane potential detection
The changes in mitochondrial membrane potential induced by ROS were characterized by a mitochondrial membrane potential assay kit. CT26 cells were seeded onto 6-well plates ($10^6$ cells for each well) and allowed to grow for 24 h at 37 °C with 5% $CO_2$. Afterward, the cells were

treated with different nanoparticles, including BST NSs, $CaO_2$ NPs, and BST/$CaO_2$ NSs, at a final concentration of 0.1 mg/mL. Following a 12-h treatment period, the CT26 cells were washed three times with PBS and stained with JC-1. After a 20-min incubation, the stained CT26 cells were washed three times with JC-1 staining buffer. Finally, CLSM was used to determine changes in mitochondrial membrane potential under different treatments.

### Live/dead cell staining experiment
A Calcein AM·PI assay kit was applied to detect the cellular state under different treatments. CT26 cells, after seeding on 6-well plates ($10^6$ cells for each well) for 24 h, were treated with 0.1 mg/mL BST NSs, $CaO_2$ NPs, and BST/$CaO_2$ NSs for 6 h. Then, the treated cells were stained with a Calcein AM·PI assay kit for 30 min. Finally, the dead cells and live cells were characterized by red fluorescence and green fluorescence using CLSM.

### Apoptosis analysis
CT26 cells were seeded onto 6-well plates ($10^6$ cells for each well) and allowed to grow for 24 h at 37 °C with 5% $CO_2$. CT26 cells were treated with 0.1 mg/mL BST NSs, $CaO_2$ NPs, and BST/$CaO_2$ NSs for 6 h. A Dead Cell Apoptosis Kit with Annexin V FITC and PI (Beyotime) was applied to stain the treated cells to determine cell apoptosis. After that, the apoptosis of CT26 cells was quantified with FCM analysis.

### Dendritic cell maturation measurement
Bone marrow cells were isolated from the femurs and tibias of 7-week-old female Balb/c mice. Then, the isolated bone marrow cells were differentiated into bone marrow-derived DCs (BMDCs). BMDCs cells were seeded onto 6-well plates ($10^6$ cells for each well) and allowed to incubate for 6 days at 37 °C with 5% $CO_2$. Afterward, the cells were incubated with 0.1 mg/mL BST NSs, $CaO_2$ NPs, and BST/$CaO_2$ NSs for 24 h and stained with anti-CD86 PE, anti-CD80 APC, and anti-CD11c FITC for FCM analysis.

### Establishment of subcutaneous xenograft colorectal cancer animal model
All animal experiments were approved by the Animal Ethics Committee of the Tianjin University Laboratory Animal Center (Tianjin, China) (Approval No. TJUE-2022-210) and carried out according to the Guidelines for the Care and Use of Laboratory Animals of Tianjin University. Here, 2000 $mm^3$ was set as the permitted maximal tumor size in conventional in vivo antitumor experiments. To further study the metastasis of advanced colorectal cancer and biosafety of different treatments, five mice in each group were euthanized on the tenth day under the premise of ensuring a regular diet, stable weight and no obvious pain. For two mice in the Control group and one mouse in the BST group, the maximal tumor size limit exceeded 2000 $mm^3$, but the experiment continued. This deviation was approved by the Ethics Committee, and mice were closely monitored. One hundred microliters of serum-free cell medium containing $2 \times 10^6$ CT26 cells was injected subcutaneously into BALB/c mice (female, 6 weeks, 14–16 g) to establish a xenograft tumor model.

### Establishment of orthotopic colorectal cancer animal model
All animal experiments were approved by the Animal Ethics Committee of the Tianjin University Laboratory Animal Center (Tianjin, China) (Approval No. TJUE-2022-210) and carried out according to the Guidelines for the Care and Use of Laboratory Animals of Tianjin University. Here, 2000 $mm^3$ was set as the permitted maximal tumor size in conventional in vivo antitumor experiments. The mice were first anesthetized by isoflurane. After being disinfected with an alcohol swab on the abdomen, the mice received a median incision on the lower abdomen to exteriorize the cecum. CT26-Luc cells ($2 \times 10^6$) were injected into the colorectum wall of BALB/c mice (female, 6 weeks, 14–16 g) with a 30-gauge needle. Then, the cecum was returned to the abdominal cavity, and the abdominal cavity was sutured with a 5-0 absorbable suture.

### Measurement of antitumor effects
**Subcutaneous xenograft colorectal cancer animal model.** The CT26 tumor-bearing mice were randomly divided into four treatment groups, and the tumors reached approximately 80 $mm^3$ with five mice each as follows: PBS, BST NSs, $CaO_2$ NPs, and BST/$CaO_2$ NSs. The BST NSs and BST/$CaO_2$ NSs were injected intravenously at a dose of 5 mg/kg. Because the loading capacity of $CaO_2$ NPs on BST/$CaO_2$ NSs was 10 wt%, the $CaO_2$ NPs were injected intravenously at a dose of 0.5 mg/kg. The tumor size of each mouse in the different groups was measured and recorded by a caliper and digital scale every 2 days during the treatment. The tumor volumes were calculated according to the following formula: tumor volume = (length × width²)/2. To further study the metastasis of advanced colorectal cancer and biosafety of different treatments, five mice in each group were euthanized on the tenth day under the premise of ensuring a regular diet, stable weight and no obvious pain. All animal experimental protocols were approved by the Animal Ethics Committee.

**Orthotopic colorectal cancer animal model.** Half a month after orthotopic tumor cell inoculation, the mice received an intraperitoneal injection of d-luciferin (150 mg kg⁻¹) to check the bioluminescence intensity of the tumor. Mice with a bioluminescence intensity of ~$1 \times 10^6$ photons (p) s⁻¹ cm⁻² sr⁻¹ were used for in vivo anticancer experiments. Then, the mice were randomly divided into four groups ($n = 5$) and received rectal perfusion of PBS, BST NSs (5 mg/kg), $CaO_2$ NPs (0.5 mg/kg), and BST/$CaO_2$ NSs (5 mg/kg) three times a week. Furthermore, the mice in each group received intraperitoneal injection of d-luciferin (150 mg kg⁻¹) after 3 weeks of treatment. The mice were anesthetized and imaged under a small animal imaging system to monitor tumor bioluminescence.

### Fluorescence imaging and biodistribution study
Cy5.5-labeled BST/$CaO_2$ NSs were injected intravenously into CT26 tumor-bearing BALB/c mice (female, 6 weeks, 14–16 g). The fluorescence of the whole body of mice was recorded by a Maestro2 in vivo imaging system. Twenty-four hours postinjection, the mice were sacrificed, and the tumors and major organs were collected and imaged. The ImageJ analysis system was applied to measure the fluorescence intensity of Cy5.5-labeled BST/$CaO_2$ NSs in major organs and tumors. Then, the intensity values were normalized using the weight (grams) of each organ and tumor.

### PA imaging in vivo
To test the PA imaging of BST/$CaO_2$ NSs, the PA signal was detected by the MSOT inVision PA imaging system (inVision 256-TF, iThera Medical). In detail, tumor-bearing BALB/c mice (female, 6 weeks, 14–16 g) were intravenously injected with BST/$CaO_2$ NSs before imaging. After 12 and 24 h, tumor-bearing mice were imaged by a small animal MSOT inVision PA imaging system.

### CT imaging in vivo
The in vivo CT imaging was carried out by a small mouse X-ray CT (Gamma Medica-Ideas). Imaging parameters were as follows: field of view, 80 by 80 mm; slice thickness, 154 μm; effective pixel size, 50 μm; tube voltage, 80 kV; tube current, 270 μA. The CT images were analyzed using amira 4.1.2. In detail, tumor-bearing BALB/c mice (female, 6 weeks, 14–16 g) were intravenously injected with BST/$CaO_2$ NSs before imaging. After 12 and 24 h, tumor-bearing mice were imaged by small animal X-ray CT. The mouse whole-body 360° scan lasted approximately 20 min under isophane anesthesia.

**Article**

## In vivo biosafety measurements

BALB/c mice (female, 6 weeks, 14–16 g) were used to investigate biosafety. To investigate the biosafety of $BST/CaO_2$ NS-based cancer therapy, the apoptosis and DNA damage of vital organs (including heart, liver, spleen, lung, and kidney) of subcutaneous xenograft colorectal cancer mice after different treatment treatments were investigated. On the tenth day of treatment, the mice were euthanized, and their vital organs and tumors were extracted for flow cytometry and H&E staining analysis. For the $BST/CaO_2$ NS-treated mice, inflammatory factors, including TNF-a, IL-8, IL-1b, IL-6, and IFN-g, were detected by real-time quantitative PCR (RT–qPCR) on the tenth day of treatment.

In addition, healthy BALB/c mice (female, 7 weeks, 16–18 g) were intravenously injected with $BST/CaO_2$ NSs at a high dose of 10 mg/kg. At 1 day, 7 days, and 14 days after injection, complete blood counts including platelet (PLT), white blood cells (WBC), mean corpuscular volume (MCV), red blood cells (RBC), mean corpuscular hemoglobin concentration (MCHC), mean corpuscular hemoglobin (MCH), hematocrit (HCT), hemoglobin (HGB), and serum biochemical parameters including γ-glutamyl transpeptidase (γ-GT), total protein (TP), creatine kinase (CK), lactate dehydrogenase (LDH), c-reactive protein (CRP), creatinine (Cr), blood urea nitrogen (BUN), alanine aspartate aminotransferase (AST), amylase (AMY), aminotransferase (ALT), and albumin (ALB) were detected and compared with the control group to estimate the biocompatibility of $BST/CaO_2$ NSs.

## Reporting summary

Further information on research design is available in the Nature Portfolio Reporting Summary linked to this article.

## Data availability

The authors declare that all data supporting the findings of this study are available within the article and the Supplementary Information. The full image dataset is available from the corresponding author upon request. Source data are provided with this paper.

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

## Acknowledgements

This study was financially supported by a grant from the National Natural Science Foundation of China (Grant No. 32071322, X.J.), National Natural Science Funds for Excellent Young Scholars (Grant No. 32122044, X.J.) and Technology & Innovation Commission of Shenzhen Municipality (Grant No. JCYJ20210324113004010, X.J.).

## Author contributions

X.J. and D.M. designed and supervised the project. X.J. and X.Y. designed the experimental strategies. X.Y., Y.K., J.D., R.L., J.Y., Y.F., J.H., and J.Y. performed the experiments and analyzed the data. X.J., X.Y., and G.N. wrote the manuscript.

## Competing interests

The authors declare no competing interests.
