## [Peer Review File · Nature Communications]

Reviewers' Comments:

Reviewer #1:

Remarks to the Author:

The study by Yuan et al. is relevant and interesting to the broad readership of the journal. The study is well organized.

However, I have doubts regarding the innovation and also there are some major revisions that need to be addressed and in vivo data that need to be provided prior to re-consider the study:

- 1) The text contains many grammar issues with errors. Please revise the language and fluency of the whole text.
- 2) In Figure 5b, the authors need to show the in vivo fluorescence images for all animals or at least 3 animals. The same for the organs. Not just 1 animal or the organs from 1 animal. Please revise this.
- 3) Regarding Figure 5f, photographs of CT26 xenograft tumor-bearing mice under different treatments, one can barely see anything with these type of images. The authors already have the measures of tumor volume in Figs. 5d and 5e. All these images can go to supporting information.
- 4) In figure 6, the authors should quantify the level of damage for the tumors and each organ. This should be added in Figure 6.
- 5) The authors need to provide a survival curve as a proxy of the therapeutic efficiency of the implants. The authors can consider a survival cutoff criteria that can include tumour ulceration or compassionate euthanasia, when the aggregate tumour burden >1 cm in diameter, or if the tumour impeded eating, urination, defecation or ambulation.
- 6) In Fig7 B and C need to add the Y axis title in the graphs.

Reviewer #2:

Remarks to the Author:

In this research article, the authors investigated an interesting approach to induce a "ROS surge" by utilizing a pH-activated thermoelectric nanosystem, namely BST/CaO₂ nanosheets (NS). The proposed mechanism involves the dissolution of the CaO₂ surface layer in response to the acidic tumor microenvironment, leading to heat generation and subsequent activation of BST to generate ROS for temperature-driven cancer therapy. Furthermore, the released Ca²⁺ ions in this process may trigger a series of tumor-limiting pathways, ultimately serving the purpose of self-triggered thermoelectric cancer/immunotherapy.

Introduction:

1. When stating "photocatalysts have very limited access to light energy in vivo" and "most of photocatalysts with wide band gap can only respond to short wavelength light", please notice that at least 11 photoimmunotherapy clinical trials have been registered on clinicaltrials.gov, and most of them are using a NIR dye named IRDye700DX.
2. The statement "High intensity or prolonged ultrasonic stimulation may cause mechanical and pathological damage to normal tissues or organs" reads a bit exaggerating and misleading. In fact, high-intensity focused ultrasound (HIFU) is a clinically verified method to treat cancer, in many hospitals, on a daily basis.
3. The proposed thermocatalytic therapy is interesting, but a quick search on PubMed showed that many two-dimensional nanosheets can elicit a similar effect. This leads to the question that why the authors chose Bi_{0.5}Sb_{1.5}Te₃ with such a strong commitment. Is there any comparison between different types of nanosystems or different composition of BiSbTe nanosheets?
4. The released Ca²⁺ ions propose another concern. Calcium homeostasis is important in maintaining stable functioning of a living organism. High level of calcium in the blood, or hypercalcemia, is known to cause damage in the kidney, bone, brain, and digestive system. For

treatment, 3 doses of the nanosystems were injected but no information was shown regarding blood levels of calcium.

Results

5.The characterization of the nanosystem is comprehensive and well-executed, reflecting the authors' outstanding expertise in the field of nanomedicine and cancer theranostics.

6.For in vivo fluorescent imaging, Cy5.5 is known to have a strong background in the stomach and GI tract, making it a less ideal fluorophore for biodistribution analysis.

7.Also, FL/gram tissue is not very convincing for biodistribution studies, ICP/MS or other more accurate quantification methods are recommended.

8.According to Fig. 5C, liver, kidney, and lung share most of the injected nanosystems. Although toxicity analysis did not reveal any significant adverse effects, the nanosystem's metabolic pathway, biological fate, and comprehensive toxicity profile still stand as the rule of thumb before utilizing it in cancer treatment. Such analysis is considered a standard practice in the development of safe and effective drug, including nanotherapeutics.

Discussion and Figures

9.Reading the manuscript, the idea of ROS surge reminds the reviewer of radiotherapy, which basically employs targeted radiation to induce ROS generation in the tumor. As radiotherapy would encounter hypoxia and tumor resistance among other issues, do the authors foresee any potential resistant from the current method? More demonstration in this regard would interest many clinicians.

10.In Fig.1, what do the gray ovals on cell membrane mean?

11.The manuscript contains many not-so-frequently-used expressions, it would benefit from a thorough review for grammatical errors and typos.

Reviewer #3:

Remarks to the Author:

This paper describes an interesting thermal triggered nano-catalyst system that generate ROS in acidic TME. Both in vitro and in vivo data support multifunctional effects of the BST/CaO₂ NSs in generating Ca²⁺ ion surge, heat, and ROS. However, some critical control experiments are missing. Background information and literature on prior work, especially prior work on CaO₂, is completely omitted. BST and CaO₂ nano-systems have been separately reported before.

Control experiments needed for critical evaluation of the reported work:

- a. Synthesis and characterization of CaO₂ NPs.
- b. Cell viability and cell death at acidic pH.
- c. Temperature change (ΔT) in culture media for CaO₂, BST, and BST/CaO₂.
- d. Intracellular concentration of Ca²⁺ ions.

Several minor points need to be addressed.

1. Figure 2: what are the values for scale bars?
2. Evidence of heat generated when BST/CaO₂ NSs are exposed to aqueous media at different pHs.
3. Figure 5: what were injected dose for each nanoparticle? What were the concentrations of Ca²⁺ in tumor and lymph nodes?

Reviewer #1 (Remarks to the Author):

The study by Yuan et al. is relevant and interesting to the broad readership of the journal. The study is well organized. However, I have doubts regarding the innovation and there are some major revisions that need to be addressed and *in vivo* data that need to be provided prior to reconsider the study:

Response: Thank you for the reviewer's positive comments. We have followed the reviewer's comments and performed additional experiments to address the points raised by the reviewer. Please see point-by-point responses below.

For the *innovation* of our strategy reported in this manuscript, although both BST NSs and CaO₂ NPs have been reported separately for tumor therapy, the intelligent combination of BST NSs and CaO₂ NPs and their synergy-derived tumor-specific self-triggered thermoelectric therapy is what makes this paper innovative. The exogenous excitation requirement and electron-hole pair recombination are the key elements limiting the application of catalytic therapies. Tumor-specific self-triggered thermoelectric catalysis based on BST/CaO₂ heterojunction combined catalytic therapy, ion interference therapy, and immunotherapy is first reported.

Central innovation:

Tumor microenvironment (TME)-specific and self-triggered thermoelectric therapy without any external stimulation is innovative and has great potential in tumor therapy.

Collaborative innovation:

(1) Upon exposure to the acidic TME, the CaO₂ NP coating hydrolyzed rapidly and released Ca²⁺, H₂O₂, and heat.

(2) Heat: The heat induced a temperature difference on BST NSs, triggering the thermoelectric effect, which pyro-generates negative and positive charges for chemical oxidation–reduction reactions and reactive oxygen species (ROS) generation. The voltage inside the thermoelectric material (BST NSs)-induced self-built-in electric field can retard electron-hole recombination, ensuring the corresponding catalytic activity and high ROS production.

(3) H₂O₂: H₂O₂ not only provides substrate (O₂) supplementation for thermoelectric catalysis but also dysregulates Ca²⁺ channels, preventing Ca²⁺ efflux.

(4) Ca²⁺: Ca²⁺ mediates calcium overload-mediated therapy, which could be aggravated by dysregulation of calcium channels by H₂O₂. Additionally, Ca²⁺ promotes DC maturation and tumor antigen presentation, thus activating the immune response and enabling immunotherapy.

Biosafety:

In a normal physiological environment, the different and mild hydrolysis pathways of CaO₂ NPs, producing Ca²⁺ and O₂ slowly and without heat, cannot trigger the thermoelectric catalysis of BST NSs, guaranteeing high biosafety to normal organs and tissues.

The innovation of our reported TME-specific and self-triggered thermoelectric therapy was reframed and emphasized in the Abstract, Introduction, and Discussion sections.

1) The text contains many grammar issues with errors. Please revise the language and fluency of the whole text.

Response: Thank you for the reviewer's comments. We have revised the WHOLE manuscript carefully and tried to avoid any grammar or syntax errors. In addition, we

have asked colleagues who are skilled in writing scientific papers in English to check the English. We believe that the language is much improved.

2) In Figure 5b, the authors need to show the *in vivo* fluorescence images for all animals or at least 3 animals. The same was true for the organs. Not just 1 animal or the organs from 1 animal. Please revise this.

Response: Thank you for this valuable comment. The *in vivo* fluorescence images of 3 mice are shown in Figure 6. In addition, PA and CT imaging were carried out for more accurate characterization of the distribution of nanomaterials *in vivo*. The relevant statements were added in our revised manuscript.

Methods section:

Fluorescence imaging and biodistribution study.

Cy5.5-labeled BST/CaO₂ NSs were injected intravenously into CT26 tumor-bearing mice. The fluorescence of the whole body of mice was recorded by a Maestro2 *in vivo* imaging system. Twenty-four hours postinjection, the mice were sacrificed, and the tumors and major organs were collected and imaged. The ImageJ analysis system was applied to measure the fluorescence intensity of Cy5.5-labeled BST/CaO₂ NSs in major organs and tumors. Then, the intensity values were normalized using the weight (grams) of each organ and tumor.

PA imaging *in vivo*.

To test the PA imaging of BST/CaO₂ NSs, the PA signal was detected by the MSOT inVision PA imaging system (inVision 256-TF, iThera Medical). In detail, tumor-bearing mice were intravenously injected with BST/CaO₂ NSs before imaging. After 12 and 24 hours, tumor-bearing mice were imaged by a small animal MSOT inVision PA imaging system.

CT imaging *in vivo*.

The *in vivo* CT imaging was carried out by a small mouse X-ray CT (Gamma Medica-Ideas). Imaging parameters were as follows: field of view, 80 mm by 80 mm; slice thickness, 154 μm; effective pixel size, 50 μm; tube voltage, 80 kV; tube current, 270 μA. The CT images were analyzed using amira 4.1.2. In detail, tumor-bearing mice were intravenously injected with BST/CaO₂ NSs before imaging. After 12 and 24 hours, tumor-bearing mice were imaged by small animal X-ray CT. The mouse whole-body 360° scan lasted approximately 20 min under isophane anesthesia.

Results section:

***In vivo* imaging and biodistribution of BST/CaO₂ NSs.**

To evaluate the *in vivo* therapeutic performance of the BST/CaO₂ NS-based self-triggered thermoelectric system, CT26 xenograft tumor models were established in BALB/c mice. To investigate the biodistribution of the Cy5.5-labeled BST/CaO₂ NSs, they were intravenously injected into CT26 xenograft tumor models prior to evaluating their antitumor effect. The biodistribution of the BST/CaO₂ NSs was observed at 4, 12, and 24 hours postinjection using *in vivo* imaging, and it was found that there was an effective and continuous accumulation of the nanoscale particles at the tumor site (Fig. 6a). This was further confirmed by semiquantitative analysis of BST/CaO₂ NSs in the major organs (including the heart, liver, spleen, lung, and kidney) and tumors 24 hours after intravenous injection. As shown in Fig. 6b, a bright fluorescence signal was present in the dissected tumor, which was in agreement with the *in vivo* imaging results. Supplementary Fig. 13 shows the semiquantitative analysis of BST/CaO₂ NSs in the major organs and tumor 24 h after intravenous injection, which was in agreement with

the *in vivo* imaging results, further demonstrating the EPR effect-induced accumulation of nanoscale BST/CaO₂ NSs at the tumor site. To more accurately characterize the distribution of the BST/CaO₂ NSs *in vivo*, photoacoustic (PA) imaging and computerized tomography (CT) were used to conduct real-time monitoring. Because of the excellent photothermal conversion performance of the BST NSs, they served as a PA indicator for *in vivo* photoacoustic imaging. Real-time PA images of the tumor-bearing mice were recorded after intravenous injection with BST/CaO₂ NSs. The findings suggest that the BST/CaO₂ NS-based self-triggered thermoelectric system has great potential for use as a synergistic antitumor therapy *in vivo* due to its effective accumulation at the tumor site. As shown in Fig. 6c, BST/CaO₂ NSs accumulated in the tumor site well over time. Furthermore, it should be noted that the BST/CaO₂ NSs also exhibit potential as CT imaging agents due to the high X-ray attenuation coefficient of Bi. In fact, as demonstrated in Fig. 6d and 6e, there is a positive correlation between the concentration of BST/CaO₂ NSs and the Hounsfield unit (HU) value, indicating their ability to serve as effective contrast agents for CT imaging. To evaluate their *in vivo* CT imaging potential, BST/CaO₂ NSs were intravenously injected into CT26 tumor-bearing mice and analyzed using coronal CT imaging. The results, displayed in Fig. 6f, showed enhanced contrast within the tumor area, suggesting the potential for BST/CaO₂ NSs to serve as efficient CT imaging agents for cancer diagnosis. Moreover, to further investigate the biodistribution of BST/CaO₂ NSs *in vivo*, ICP/MS analysis was utilized, as depicted in Supplementary Fig. 14. The results indicated a significant accumulation of NSs within the major organs and tumors over a period of 30 days, highlighting their effectiveness in targeting tumors. Importantly, Supplementary Fig. 14 also illustrates that the accumulated BST/CaO₂ NSs within normal organs and tissues were gradually excreted by the body over time, indicating their biocompatibility and potential for clinical translation.

Fig. 6. *In vivo* imaging and biodistribution of the BST/CaO₂ heterojunction. a *In vivo* fluorescence images of tumor-bearing mice at different time points after intravenous injection with Cy5.5-labeled BST/CaO₂ heterojunction and **b** *ex vivo* fluorescence images of tumor and major organs at 24 hours after injection. **c** *In vivo* photoacoustic images of tumor-bearing mice after intravenous injection with BST/CaO₂ heterojunction. **d** CT images of BST/CaO₂ heterojunctions with different concentrations. **e** The CT values (HU) of the BST/CaO₂ heterojunction. **f** *In vivo* CT

images of tumor-bearing mice after intravenous injection with the BST/CaO₂ heterojunction. The red circle indicates a tumor.

Supplementary Figure 13. Semiquantitative analysis of the biodistribution of Cy5.5-labeled BST/CaO₂ NSs in CT26 xenograft tumor-bearing mice.

Supplementary Figure 14. Biodistribution of BST/CaO₂ NSs at different times.

3) Regarding Figure 5f, photographs of CT26 xenograft tumor-bearing mice under different treatments, one can barely see anything with these type of images. The authors already have the measures of tumor volume in Figs. 5d and 5e. All these images can go to supporting information.

Response: Thank you for your comments. As suggested by the reviewer, Figure 5f has been removed. In addition, an orthotopic colorectal cancer animal model was established by injecting CT26-luc cells (2×10^6) into the cecal wall of mice to further evaluate the antitumor effect of the BST/CaO₂ NS-based self-triggered thermoelectric strategy.

Methods section:

Measurement of antitumor effects.

Subcutaneous xenograft colorectal cancer animal model:

The CT26 tumor-bearing mice were randomly divided into four treatment groups, and the tumors reached approximately 80 mm³ with five mice each as follows: PBS, BST NSs, CaO₂ NPs, and BST/CaO₂ NSs. The BST NSs and BST/CaO₂ NSs were injected intravenously at a dose of 5 mg/kg. Because the loading capacity of CaO₂ NPs on BST/CaO₂ NSs was 10 wt%, the CaO₂ NPs were injected intravenously at a dose of 0.5 mg/kg. The body weight and tumor size of each mouse in the different groups were measured and recorded by a caliper and digital scale every 2 days during the treatment. The tumor volumes were calculated according to the following formula: tumor volume = (length × width²)/2.

Orthotopic colorectal cancer animal model:

Half a month after orthotopic tumor cell inoculation, the mice received intraperitoneal injection of d-luciferin (150 mg kg⁻¹) to check the bioluminescence intensity of the tumor. Mice with a bioluminescence intensity of $\sim 1 \times 10^6$ photons (p) s⁻¹ cm⁻² sr⁻¹ were used for *in vivo* anticancer experiments. Then, the mice were randomly divided into four groups ($n = 5$) and received rectal perfusion of PBS, BST NSs (5 mg/kg), CaO₂ NPs (0.5 mg/kg), and BST/CaO₂ NSs (5 mg/kg) three times a week. Furthermore, the mice in each group received intraperitoneal injection of d-luciferin (150 mg kg⁻¹) after three weeks of treatment. The mice were anesthetized and imaged under a small animal imaging system to monitor tumor bioluminescence.

Results section:

Additionally, an orthotopic colorectal cancer animal model was established by injecting CT26-luc cells (2×10^6) into the colorectal wall of mice to evaluate the antitumor effect of the BST/CaO₂ NS-based self-triggered thermoelectric strategy. Once the bioluminescence intensity of the colon reached 1×10^6 photons (p) s⁻¹ cm⁻² sr⁻¹, the mice were randomly divided into four groups ($n = 5$) and received rectal perfusion of PBS, BST NSs, CaO₂ NPs, or BST/CaO₂ NSs three times a week. Three weeks later, the mice in each group received intraperitoneal injection of d-luciferin (150 mg kg⁻¹) and were anesthetized and imaged under the small animal imaging system to monitor tumor bioluminescence. As shown in Fig. 7d, the representative bioluminescence images demonstrated that BST NSs alone did not delay tumor growth compared to the control group. However, the tumor bioluminescence change images and curves in Fig. 7e revealed that CaO₂ NPs exhibited moderate *in vivo* anticancer effects during treatment. Encouragingly, BST/CaO₂ NSs showed a marked inhibitory effect on tumor growth compared to saline, demonstrating the high antitumor efficiency of the BST/CaO₂ NS-based self-triggered thermoelectric strategy in the orthotopic colorectal cancer animal model.

Fig. 7. *In vivo* antitumor performance of the self-triggered thermoelectric system. **a** Experimental illustration of *in vivo* antitumor therapy. **b, c** Antitumor performance of different treatments, including control, BST, CaO₂ and BST/CaO₂, on a subcutaneous xenograft colorectal cancer animal model. Data are presented as the mean \pm s.d. ($n = 5$ biologically independent mice). Statistical differences were analyzed by Student's two-sided t test. **d, e** Antitumor performance of different treatments, including control, BST, CaO₂ and BST/CaO₂, on an orthotopic colorectal cancer animal model. Data are

presented as the mean \pm s.d. (n = 5 biologically independent mice). Statistical differences were analyzed by Student's two-sided t test. **f** The survival curves of tumor-bearing mice under different treatments. **g, h** Flow cytometry analysis of the percentage of DC maturation and migration to lymph nodes (CD11c⁺ CD80⁺ CD86⁺) under different treatments. Data are presented as the mean \pm s.d. (n = 5 biologically independent mice). Statistical differences were analyzed by Student's two-sided t test.

4) In figure 6, the authors should quantify the level of damage for the tumors and each organ. This should be added in Figure 6.

Response: The reviewer's comment is very constructive and useful. The level of damage to the tumors and each organ after treatment with BST/CaO₂ NSs was quantified by ImageJ and added to our revised manuscript. Moreover, to quantitatively measure the apoptosis and DNA damage of organs and tumors in each treatment more accurately, we used flow cytometry to conduct detailed quantitative analysis of organs and tumors of mice in different treatment groups. The results and relevant statements have been added to our revised manuscript.

Results section:

To further investigate the underlying mechanism, tumor sections treated with BST/CaO₂ NSs were analyzed using immunofluorescence (IF) staining of γ -H2AX and cleaved caspase-3 (C-CAS3) as markers for DNA double-strand breakage and cell apoptosis, respectively. As shown in Fig. 8a and Supplementary Fig. 21, high levels of irreparable DNA damage and cell apoptosis were observed in tumor sections treated with BST/CaO₂ NSs due to the self-triggered thermoelectric catalysis and Ca²⁺-induced immunoregulation of the particles. Additionally, the self-triggered thermoelectric and immunotherapy of BST/CaO₂ NSs was confirmed through TUNEL staining, which revealed a larger area of apoptosis in cancer cells after treatment with BST/CaO₂ NSs (Fig. 8c). These findings demonstrate the efficient and synergistic effects of the self-triggered thermoelectric and immunotherapy of BST/CaO₂ NSs. Furthermore, to obtain a more accurate quantitative measurement of apoptosis and DNA damage in organs and tumors in each treatment group, we used FCM to conduct detailed analysis. As depicted in Fig. 9 and Supplementary Figs. 22 and 23, no significant DNA damage or apoptosis was observed in normal organs (heart, liver, spleen, lung, kidney) across all treatment groups. However, BST/CaO₂ NSs showed significant DNA damage and apoptosis in tumor tissues, confirming the tumor specificity and biosafety of the self-triggered thermoelectric strategy based on BST/CaO₂ NSs.

Apoptosis

DNA damage

Fig. 9. Quantitative analysis of apoptosis and DNA damage in major organs (heart, liver, spleen, lung, and kidney) and tumors under different treatments by FCM. a Apoptosis in the heart, liver, spleen, lung, kidney, and tumor under different treatments. **b** DNA damage in the heart, liver, spleen, lung, kidney, and tumor under different treatments.

Supplementary Figure 21. Quantitatively measure the apoptosis and DNA damage of organs and tumors after treatment with BST/CaO₂.

5) The authors need to provide a survival curve as a proxy of the therapeutic efficiency of the implants. The authors can consider a survival cutoff criteria that can include tumor ulceration or compassionate euthanasia, when the aggregate tumor burden >1 cm in diameter, or if the tumor impeded eating, urination, defecation or ambulation.

Response: Thank you for this valuable comment. The survival curve has been added in our revised manuscript.

Fig. 7. f The survival curves of tumor-bearing mice under different treatments.

6) In Fig 7 B and C need to add the Y axis title in the graphs.

Response: Thank you for your helpful comments. The Y-axis title in Fig 7B and C has been added in our revised manuscript.

Fig. 10. Biosafety assessment of BST/CaO₂ NS-based self-triggered thermoelectric and immunological therapy. **a** Representative H&E-stained images of the heart, liver, lung, kidney, and spleen under different treatments. Scale bars = 300 μ m. **b** Analysis of inflammatory factors in the heart, liver, lung, kidney, and spleen under different treatments. Data are presented as the mean \pm s.d. ($n = 5$ biologically independent mice). Statistical differences were analyzed by Student's two-sided t test. **c** Blood hematology analysis of Balb/c mice under different treatments. Data are presented as the mean \pm s.d. ($n = 5$ biologically independent mice). Statistical differences were analyzed by Student's two-sided t test. **d** Blood biochemical analysis of Balb/c mice under different treatments. Data are presented as the mean \pm s.d. ($n = 5$ biologically independent mice). Statistical differences were analyzed by Student's two-sided t test.

Reviewer #2 (Remarks to the Author):

In this research article, the authors investigated an interesting approach to induce a “ROS surge” by utilizing a pH-activated thermoelectric nanosystem, namely BST/CaO₂ nanosheets (NS). The proposed mechanism involves the dissolution of the CaO₂ surface layer in response to the acidic tumor microenvironment, leading to heat generation and subsequent activation of BST to generate ROS for temperature-driven cancer therapy. Furthermore, the released Ca²⁺ ions in this process may trigger a series of tumor-limiting pathways, ultimately serving the purpose of self-triggered thermoelectric cancer/immunotherapy.

Response: We very much appreciate the reviewer’s thoughtful and helpful comments. During the past two months, we have performed a series of additional experiments to acquire more significant data. All these data have been added accordingly to the revised manuscript. Moreover, we have also made a series of modifications/corrections/additions to the manuscript. We hope that this revised version can now address all the concerns raised by the respected reviewer and satisfy the high publication standard in *Nature Communications*. Below, please also find our point-by-point responses.

Introduction:

1. When stating “photocatalysts have very limited access to light energy *in vivo*” and “most of photocatalysts with wide band gap can only respond to short wavelength light”, please notice that at least 11 photoimmunotherapy clinical trials have been registered on clinicaltrials.gov, and most of them are using a NIR dye named IRDye700DX.

Response: Thank you very much for this valuable comment. The inaccurate statement has been revised.

Introduction section:

Although photocatalytic therapy is capable of converting light energy into chemical energy, its practical application is limited by various factors.^{10, 16-22} Therefore, alternative approaches such as piezocatalytic therapy have been explored to address these limitations and improve cancer treatment outcomes. In photocatalysis, light irradiation is necessary to initiate the process. However, this requirement adds complexity and limits its effectiveness since traditional photocatalysts have limited access to visible light energy in living organisms due to the barrier function of skin and other biological tissues. This scarcity of extrinsic light energy results in a limitation of the photocatalytic process.^{17, 22} To meet biomedical requirements, the fast recombination of photoexcited electron-hole pairs in both the surface and bulk phases of photocatalysts needs to be minimized.¹¹ This has been a significant challenge for photocatalytic therapy. However, piezocatalytic therapy based on the piezoelectric effect offers an alternative approach. By generating a piezoelectric potential, this therapy can drive charge separation or transfer and trigger redox reactions.²³⁻²⁷ Essentially, this converts mechanical energy into chemical energy, but it does require the use of an additional external force, such as an ultrasonic generator.

2. The statement “High intensity or prolonged ultrasonic stimulation may cause mechanical and pathological damage to normal tissues or organs” reads a bit exaggerating and misleading. In fact, high-intensity focused ultrasound (HIFU) is a clinically verified method to treat cancer in many hospitals on a daily basis.

Response: We appreciate this helpful comment. The inaccurate statement has been deleted in our revised manuscript.

3. The proposed thermocatalytic therapy is interesting, but a quick search on PubMed showed that many two-dimensional nanosheets can elicit a similar effect. This leads to the question that why the authors chose $\text{Bi}_{0.5}\text{Sb}_{1.5}\text{Te}_3$ with such a strong commitment. Is there any comparison between different types of nanosystems or different composition of BiSbTe nanosheets?

Response: We thank the reviewer for this helpful comment. Although there are many thermoelectric materials with excellent properties, the thermoelectric materials in self-triggered thermoelectric therapy need to meet the following three conditions: first, a high Seebeck coefficient, high electrical conductivity, and low thermal conductivity are the basic conditions for an excellent thermoelectric material. Second, most prepared thermoelectric materials with extremely high thermoelectric conversion require very high temperature. However, *in vivo* applications require thermoelectric materials with high conversion efficiency at relatively low temperatures. Finally, a simple and economical synthesis strategy is also necessary. Based on the above conditions, we choose thermoelectric materials such as $\text{Bi}_x\text{Sb}_{2-x}\text{Te}_3$ as the research object. The effect of different compositions on the thermoelectric conversion efficiency of $\text{Bi}_x\text{Sb}_{2-x}\text{Te}_3$ has been reported (Nano Energy, 2016, 20, 144–155), and $\text{Bi}_{0.5}\text{Sb}_{1.5}\text{Te}_3$ has the highest thermoelectric conversion efficiency at lower temperatures. In addition, the choice of thermoelectric materials is not truly the focus and innovation of this research. The concept of TME self-triggering thermoelectric therapy is what we want to share. The relevant statements have been added in our revised manuscript.

Introduction section:

To enable biomedical applications *in vivo*, it is essential to have thermoelectric materials with high conversion efficiency at low temperatures. $\text{Bi}_x\text{Sb}_{2-x}\text{Te}_3$ NSs exhibit a high Seebeck coefficient, high electrical conductivity, low thermal conductivity, and high thermoelectric conversion, making them suitable for self-triggered thermoelectric cancer therapy.^{37,44} $\text{Bi}_{0.5}\text{Sb}_{1.5}\text{Te}_3$ nanosheets (BST NSs) have been found to have the highest thermoelectric conversion efficiency at very low temperatures, making them ideal for biomedical applications *in vivo*.⁴⁴ Additionally, CaO_2 nanoparticles serve as a reservoir of calcium ions (Ca^{2+}) and hydrogen peroxide (H_2O_2) and have shown promising results in calcium overload-mediated therapy.⁴⁵⁻⁴⁷ However, CaO_2 NPs also have the potential to act as an *in vivo* switch for self-triggering thermoelectric therapy due to their tumor microenvironment (low pH)-specific water liberation thermal effect, which has not yet been explored.

44. Hong, M., Chen, Z. G., Yang, L., Zou, J. $\text{Bi}_x\text{Sb}_{2-x}\text{Te}_3$ nanoplates with enhanced thermoelectric performance due to sufficiently decoupled electronic transport properties and strong wide-frequency phonon scatterings. *Nano Energy* **20**, 144-155 (2016).
45. Zhang, M., Song, R., Liu, Y., Yi, Z., Meng, X., Zhang, J., Tang, Z., Yao, Z., Liu, Y., Liu, X., Bu, W. Calcium-Overload-Mediated Tumor Therapy by Calcium Peroxide Nanoparticles. *Chem* **5**, 2171-2182 (2019).
46. Bai, S., Lan, Y., Fu, S., Cheng, H., Lu, Z., Liu, G. Connecting Calcium-Based Nanomaterials and Cancer: From Diagnosis to Therapy. *Nanomicro. Lett.* **14**, 145 (2022).
47. Guo, D., Dai, X., Liu, K., Liu, Y., Wu, J., Wang, K., Jiang, S., Sun, F., Wang, L.,

Guo, B., Yang, D., Huang, L. A Self-Reinforcing Nanoplatfom for Highly Effective Synergistic Targeted Combinatory Calcium-Overload and Photodynamic Therapy of Cancer. *Adv. Healthc. Mater.* e2202424 (2023).

4. The released Ca^{2+} ions propose another concern. Calcium homeostasis is important in maintaining the stable functioning of a living organism. High levels of calcium in the blood, or hypercalcemia, are known to cause damage to the kidney, bone, brain, and digestive system. For treatment, 3 doses of the nanosystems were injected, but no information was shown regarding blood levels of calcium.

Response: Thank you for this comment. In this research, we proposed a tumor microenvironment (TME)-responsive self-triggered thermoelectric therapy. When exposed to only the acidic TME, the CaO_2 NP coating hydrolyzed rapidly and released Ca^{2+} , H_2O_2 , and heat. Ca^{2+} released from the TME will induce ion interference therapy in tumor cells, which will be phagocytosed by DCs after tumor cell fragmentation and induce their maturation. The CaO_2 NP coating hydrolyzed very slowly in normal cells to generate Ca^{2+} and O_2 , in which the slowly released Ca^{2+} was expelled from the cells *via* calcium channel proteins. Therefore, the relatively high Ca^{2+} concentration only occurred in tumor cells and DCs. Moreover, the mice were given a very low dose of 0.5 mg/kg CaO_2 NPs, in which approximately 10 μg of CaO_2 NPs was injected into each mouse.

The concentrations of Ca^{2+} in blood, tumor cells, and lymphocytes were detected and added to our revised manuscript. Moreover, the biosafety of BST/ CaO_2 NS-based cancer therapy has been comprehensively studied in our revised manuscript.

Results section:

Although three doses of BST/ CaO_2 NSs were injected into each mouse, the concentrations of both BST NSs and CaO_2 NPs were very low, at 0.1 and 0.01 mg per mouse, respectively. The concentrations of Ca^{2+} in blood exhibited no significant fluctuation due to the slow hydrolysis of CaO_2 NPs under normal conditions (Supplementary Fig. 15).

To further confirm the Ca^{2+} -mediated ion interference therapy, the concentration of Ca^{2+} in tumor cells was measured. The results demonstrated that each injection increased the Ca^{2+} concentration in the subcutaneous graft tumor cells, which gradually returned to normal levels over time (Supplementary Fig. 15).

To confirm the role of Ca^{2+} in promoting the maturation of DC cells, we measured the concentration of Ca^{2+} in lymphocytes at lymph nodes. As depicted in Supplementary Fig. 15, the concentration of Ca^{2+} in lymphocytes was positively correlated with the number of injections, indicating that an increase in Ca^{2+} concentration could be detected in lymphocytes 12 hours after each injection. However, over time, the concentration of Ca^{2+} in the lymphocytes gradually returned to normal levels.

Supplementary Figure 15. The concentrations of Ca^{2+} in blood, tumor cells, and lymphocytes at different times under treatment with BST/ CaO_2 NSs. The black arrow represents the three injection times.

Results

5. The characterization of the nanosystem is comprehensive and well executed, reflecting the authors' outstanding expertise in the field of nanomedicine and cancer theranostics.

Response: Thank you for this comment. We very much appreciate it.

6. For *in vivo* fluorescent imaging, Cy5.5 is known to have a strong background in the stomach and GI tract, making it a less ideal fluorophore for biodistribution analysis.

Response: Thank you very much for this valuable comment. Although Cy5.5 is known to have a strong background in the stomach and GI tract, numerous studies have used Cy5.5 as a fluorescent agent for fluorescence imaging. In our revised manuscript, to more accurately characterize the distribution of BST/ CaO_2 NSs *in vivo*, photoacoustic (PA) imaging and computerized tomography (CT) were further used to conduct real-time monitoring of the distribution of BST/ CaO_2 NSs *in vivo*.

Methods section:

Fluorescence imaging and biodistribution study.

Cy5.5-labeled BST/ CaO_2 NSs were injected intravenously into CT26 tumor-bearing mice. The fluorescence of the whole body of mice was recorded by a Maestro2 *in vivo* imaging system. Twenty-four hours postinjection, the mice were sacrificed, and the tumors and major organs were collected and imaged. The ImageJ analysis system was applied to measure the fluorescence intensity of Cy5.5-labeled BST/ CaO_2 NSs in major organs and tumors. Then, the intensity values were normalized using the weight (grams) of each organ and tumor.

PA imaging *in vivo*.

To test the PA imaging of BST/ CaO_2 NSs, the PA signal was detected by the MSOT inVision PA imaging system (inVision 256-TF, iThera Medical). In detail, tumor-bearing mice were intravenously injected with BST/ CaO_2 NSs before imaging. After

12 and 24 hours, tumor-bearing mice were imaged by a small animal MSOT inVision PA imaging system.

CT imaging *in vivo*.

The *in vivo* CT imaging was carried out by a small mouse X-ray CT (Gamma Medica-Ideas). Imaging parameters were as follows: field of view, 80 mm by 80 mm; slice thickness, 154 μm ; effective pixel size, 50 μm ; tube voltage, 80 kV; tube current, 270 μA . The CT images were analyzed using amira 4.1.2. In detail, tumor-bearing mice were intravenously injected with BST/CaO₂ NSs before imaging. After 12 and 24 hours, tumor-bearing mice were imaged by small animal X-ray CT. The mouse whole-body 360° scan lasted approximately 20 min under isophane anesthesia.

Results section:

***In vivo* imaging and biodistribution of BST/CaO₂ NSs.**

To evaluate the *in vivo* therapeutic performance of the BST/CaO₂ NS-based self-triggered thermoelectric system, CT26 xenograft tumor models were established in BALB/c mice. To investigate the biodistribution of the Cy5.5-labeled BST/CaO₂ NSs, they were intravenously injected into CT26 xenograft tumor models prior to evaluating their antitumor effect. The biodistribution of the BST/CaO₂ NSs was observed at 4, 12, and 24 hours postinjection using *in vivo* imaging, and it was found that there was an effective and continuous accumulation of the nanoscale particles at the tumor site (Fig. 6a). This was further confirmed by semiquantitative analysis of BST/CaO₂ NSs in the major organs (including the heart, liver, spleen, lung, and kidney) and tumors 24 hours after intravenous injection. As shown in Fig. 6b, a bright fluorescence signal was present in the dissected tumor, which was in agreement with the *in vivo* imaging results. Supplementary Fig. 13 shows the semiquantitative analysis of BST/CaO₂ NSs in the major organs and tumor 24 h after intravenous injection, which was in agreement with the *in vivo* imaging results, further demonstrating the EPR effect-induced accumulation of nanoscale BST/CaO₂ NSs at the tumor site. To more accurately characterize the distribution of the BST/CaO₂ NSs *in vivo*, photoacoustic (PA) imaging and computerized tomography (CT) were used to conduct real-time monitoring. Because of the excellent photothermal conversion performance of the BST NSs, they served as a PA indicator for *in vivo* photoacoustic imaging. Real-time PA images of the tumor-bearing mice were recorded after intravenous injection with BST/CaO₂ NSs. The findings suggest that the BST/CaO₂ NS-based self-triggered thermoelectric system has great potential for use as a synergistic antitumor therapy *in vivo* due to its effective accumulation at the tumor site. As shown in Fig. 6c, BST/CaO₂ NSs accumulated in the tumor site well over time. Furthermore, it should be noted that the BST/CaO₂ NSs also exhibit potential as CT imaging agents due to the high X-ray attenuation coefficient of Bi. In fact, as demonstrated in Fig. 6d and 6e, there is a positive correlation between the concentration of BST/CaO₂ NSs and the Hounsfield unit (HU) value, indicating their ability to serve as effective contrast agents for CT imaging. To evaluate their *in vivo* CT imaging potential, BST/CaO₂ NSs were intravenously injected into CT26 tumor-bearing mice and analyzed using coronal CT imaging. The results, displayed in Fig. 6f, showed enhanced contrast within the tumor area, suggesting the potential for BST/CaO₂ NSs to serve as efficient CT imaging agents for cancer diagnosis. Moreover, to further investigate the biodistribution of BST/CaO₂ NSs *in vivo*, ICP/MS analysis was utilized, as depicted in Supplementary Fig. 14. The results indicated a significant accumulation of NSs within the major organs and tumors over a period of 30 days, highlighting their effectiveness in targeting tumors. Importantly, Supplementary Fig. 14 also illustrates that the accumulated BST/CaO₂ NSs within normal organs and

tissues were gradually excreted by the body over time, indicating their biocompatibility and potential for clinical translation.

Fig. 6. *In vivo* imaging and biodistribution of the BST/CaO₂ heterojunction. a *In vivo* fluorescence images of tumor-bearing mice at different time points after intravenous injection with Cy5.5-labeled BST/CaO₂ heterojunction and **b** *ex vivo* fluorescence images of tumor and major organs at 24 hours after injection. **c** *In vivo* photoacoustic images of tumor-bearing mice after intravenous injection with

BST/CaO₂ heterojunction. **d** CT images of BST/CaO₂ heterojunctions with different concentrations. **e** The CT values (HU) of the BST/CaO₂ heterojunction. **f** *In vivo* CT images of tumor-bearing mice after intravenous injection with the BST/CaO₂ heterojunction. The red circle indicates a tumor.

Supplementary Figure 13. Semiquantitative analysis of the biodistribution of Cy5.5-labeled BST/CaO₂ NSs in CT26 xenograft tumor-bearing mice.

Supplementary Figure 14. Biodistribution of BST/CaO₂ NSs at different times.

7. Also, FL/gram tissue is not very convincing for biodistribution studies, ICP/MS or other more accurate quantification methods are recommended.

Response: We appreciate this helpful comment. ICP/MS was used to quantify the biodistribution of BST/CaO₂ NSs and added to our revised manuscript.

Supplementary Figure 14. Biodistribution of BST/CaO₂ NSs at different times.

8. According to Fig. 5C, the liver, kidney, and lung share most of the injected nanosystems. Although toxicity analysis did not reveal any significant adverse effects, the nanosystem's metabolic pathway, biological fate, and comprehensive toxicity profile still stand as the rule of thumb before utilizing it in cancer treatment. Such analysis is considered a standard practice in the development of safe and effective drugs, including nanotherapeutics.

Response: We thank the reviewer very much for these professional comments. The long-term biological fate and possible metabolic pathway of nanomaterials were detected by analyzing urinary excretion and fecal excretion by ICP/MS. The comprehensive toxicity profile of the BST/CaO₂ NS-based self-triggered thermoelectric strategy was further quantitatively tested by evaluating the levels of inflammatory factors, DNA damage, and cell apoptosis in each important organ (heart, liver, spleen, lung, and kidney) using quantitative polymerase chain reaction (qPCR) and flow cytometry (FCM). All these results exhibited negligible adverse effects, which have also been added to our revised manuscript.

Results section:

***In vivo* biosafety evaluation of BST/CaO₂ NSs.**

The *in vivo* toxicity of nanomedicine is a crucial factor for translation from bench to practical applications. Therefore, we meticulously investigated the toxicity of BST/CaO₂ NSs through histology examination, hematology assay, and immune analysis. Although three doses of BST/CaO₂ NSs were injected into each mouse, the concentrations of both BST NSs and CaO₂ NPs were very low, at 0.1 and 0.01 mg per mouse, respectively. The concentrations of Ca²⁺ in blood exhibited no significant fluctuation due to the slow hydrolysis of CaO₂ NPs under normal conditions (Supplementary Fig. 15). As shown in Fig. 6 and Supplementary Fig. 14, BST/CaO₂ NSs were distributed to a certain extent in the liver, spleen, lung, and other important organs, with their distribution decreasing over time. We hypothesize that the BST/CaO₂ NSs will be partially excreted through renal urination and intestinal defecation, which was confirmed by ICP/MS analysis of urinary and fecal excretion (Supplementary Fig. 24). This alleviated the long-term retention of BST/CaO₂ NSs in the body to some extent. The IF staining of major organs, such as the heart, liver, spleen, lung, and kidney, of BST/CaO₂ NS-treated mice with γ -H2AX and C-CAS3 as markers of DNA double-strand breaks and cell apoptosis indicated no significant apoptosis or DNA damage in

these normal organs (Fig. 8b and 8d). Moreover, TUNEL staining of normal organs of mice treated with BST/CaO₂ NSs confirmed their biosafety as an antitumor strategy. Furthermore, FCM analysis revealed no detectable DNA damage and apoptosis in the heart, liver, spleen, lung, and kidney for each treatment group compared to obvious DNA damage and apoptosis in the tumors (Fig. 9). Overall, these results suggest the biosafety of BST/CaO₂ NS-based antitumor strategies.

Hematoxylin and eosin (H&E) staining of major organs after treatment with BST/CaO₂ NSs was carried out to further confirm the biocompatibility and specific targeted antitumor mechanism of BST/CaO₂ NS-based therapy. As depicted in Fig. 10a, although intravenously injected BST/CaO₂ NSs partially accumulated in normal organs (mainly in liver, spleen, and lung), they caused almost no damage. Moreover, real-time quantitative PCR (RT-qPCR) was applied to detect the damage and inflammatory response of each major organ exposed to BST/CaO₂ NSs. The results presented in Fig. 10b confirmed the good biocompatibility and biosafety of BST/CaO₂ NS-based cancer therapy. We also performed hematological detection to investigate the systematic biosafety properties of BST/CaO₂ NSs. The mean corpuscular hemoglobin concentration (MCHC), red blood cells (RBCs), hematocrit (HCT), white blood cells (WBCs), platelets (PLTs), mean corpuscular volume (MCV), hemoglobin (HGB), and mean corpuscular hemoglobin (MCH) were measured (Fig. 10c). There was no statistically significant difference observed in the BST/CaO₂ NS-treated groups 7 and 14 days after i.v. injection compared to the control group. Furthermore, blood biochemical parameters, including γ -glutamyl transpeptidase (γ -GT), total protein (TP), C-reactive protein (CRP), creatine kinase (CK), lactate dehydrogenase (LDH), creatinine (Cr), blood urea nitrogen (BUN), alanine aspartate aminotransferase (AST), amylase (AMY), aminotransferase (ALT), and albumin (ALB), between the control mice and the mice injected with BST/CaO₂ NSs for 1, 7, and 14 days were tested. As presented in Fig. 10d, there were nearly no observable differences between the BST/CaO₂ NSs and control groups. Therefore, all of the aforementioned results demonstrate that our prepared BST/CaO₂ NSs should be considered a relatively biosafe and biocompatible nanomedicine.

Fig. 8. *In vivo* immunofluorescence staining and tissue damage analysis. a, b Immunofluorescence images of the tumors and major organs (heart, liver, spleen, lung, and kidney) obtained from mice injected with BST/CaO₂ NSs. The nucleus was stained with DAPI (blue), the damaged DNA was stained with γ -H2AX foci (red), and the apoptotic cells were stained using the apoptosis marker C-CAS3 (green). Scale bars: 1000 μ m for the first line and 100 μ m for the second and third lines. **c, d** TUNEL staining of the tumors and major organs (heart, liver, spleen, lung, and kidney) obtained from mice injected with BST/CaO₂ NSs. Scale bars: 1000 μ m for the first line and 100 μ m for the second and third lines.

Apoptosis

DNA damage

Fig. 9. Quantitative analysis of apoptosis and DNA damage in major organs (heart, liver, spleen, lung, and kidney) and tumors under different treatments by FCM. a Apoptosis in the heart, liver, spleen, lung, kidney, and tumor under different treatments. **b** DNA damage in the heart, liver, spleen, lung, kidney, and tumor under different treatments.

Fig. 10. Biosafety assessment of BST/CaO₂ NS-based self-triggered thermoelectric and immunological therapy. **a** Representative H&E-stained images of the heart, liver, lung, kidney, and spleen under different treatments. Scale bars = 300 μ m. **b** Analysis of inflammatory factors in the heart, liver, lung, kidney, and spleen under different treatments. Data are presented as the mean \pm s.d. ($n = 5$ biologically independent mice). Statistical differences were analyzed by Student's two-sided t test. **c** Blood hematology analysis of Balb/c mice under different treatments. Data are presented as the mean \pm s.d. ($n = 5$ biologically independent mice). Statistical differences were analyzed by Student's two-sided t test. **d** Blood biochemical analysis of Balb/c mice under different treatments. Data are presented as the mean \pm s.d. ($n = 5$ biologically independent mice). Statistical differences were analyzed by Student's two-sided t test.

Supplementary Figure 21. Quantitatively measure the apoptosis and DNA damage of organs and tumors after treatment with BST/CaO₂.

Supplementary Figure 24. Urinary excretion and fecal excretion of BST/CaO₂ NSs at different time points.

Discussion and Figures

9. Reading the manuscript, the idea of ROS surge reminds the reviewer of radiotherapy, which basically employs targeted radiation to induce ROS generation in the tumor. As radiotherapy would encounter hypoxia and tumor resistance among other issues, do the authors foresee any potential resistant from the current method? More demonstration in this regard would interest many clinicians.

Response: We thank the reviewer for this professional and helpful comment. The advantages and potential resistance of the current method have been analyzed and added to the Discussion section.

Discussion section:

Discussion

Catalytic therapies are a promising approach to cancer treatment, utilizing nanocatalysts that are nontoxic or low toxic to convert intracellular O₂ or H₂O into ROS such as ·O₂⁻, ·OH, and H₂O₂. These ROS induce effective tumor-specific oxidative damage and apoptosis without causing significant toxicity to normal organs or tissues.^{1, 11, 12} However, most catalytic therapies rely on exogenous excitation, which means they require specific external stimuli such as light or ultrasound to trigger catalytic reactions. Unfortunately, exogenous excitation catalytic therapy faces several challenges in clinical application. First, the penetration of light is limited, and high-intensity ultrasound may cause collateral mechanical damage. Second, the catalytic efficiency of these therapies is often low due to the fast recombination of excited holes and electrons. Last, the use of additional excitation equipment can add operational complexity and inconvenience to the treatment process.^{11, 51, 52}

Recently, a new type of catalytic therapy has been developed for cancer treatment that combines thermoelectric effects and redox reactions.^{37, 38} This approach is different from photocatalysis and piezocatalysis because it uses temperature fluctuation to generate pyro-generated negative and positive charges, which can trigger chemical oxidation–reduction reactions. Thermoelectric catalysis involves the creation of a self-built-in electric field inside a thermoelectric catalyst, which retards electron-hole recombination and allows for greater catalytic activity and higher ROS generation.⁴³ However, current thermoelectric catalysis is triggered by laser irradiation through photothermal conversion, which limits its penetration into biological tissue and reduces its catalytic efficiency. To address this issue, there is a need to develop self-triggered thermoelectric catalytic materials or systems that maintain the advantages of thermoelectric catalysis while avoiding its limitations. Such developments hold great promise for clinical applications.

In this study, we have presented a novel self-triggered thermoelectric nanoheterojunction for enhanced tumor catalytic therapy and coupling with immunotherapy. We synthesized a conventional and efficient thermoelectric biomaterial, BST NSs, and selected it as the thermoelectric catalyst. The innovation lies in the *in situ* coating of CaO₂ NPs, which not only acted as a trigger in response to the TME but also activated the immune system and imported immunotherapy. We explain that the CaO₂ NPs were hydrolyzed rapidly into Ca²⁺ and H₂O₂ in the acidic TME, generating a large amount of heat. The thermoelectric effect of BST NSs was activated by heat, producing negative and positive charges for chemical oxidation–reduction reactions and ROS generation. The self-built-in electric field inside BST NSs guided the separation of electron-hole pairs and retarded electron-hole recombination. H₂O₂ provided substrate (O₂) supplementation for thermoelectric catalysis and ROS production and regulated Ca²⁺ channels while delaying Ca²⁺ efflux. The main hydrolysate Ca²⁺ could mediate ion interference therapy, breaking intracellular ionic homeostasis and increasing the osmotic pressure of tumor cells. Additionally, it could promote DC maturation and tumor antigen presentation, thus activating an immune response and mediating effective immunotherapy. Moreover, the CaO₂ NP coating hydrolyzed slowly in normal cells to generate Ca²⁺ and O₂, where the slowly released Ca²⁺ was expelled from the cells *via* calcium channel proteins. Without the trigger of temperature fluctuation, the BST NSs possessed excellent biosafety and biocompatibility in normal organs and tissues. Overall, this study provides an intelligent strategy for the synthesis of a tumor-specific self-triggered thermoelectric catalyst and provides new insights into an advanced strategy to enhance the application scope and efficiency of catalytic therapy. Tumor-specific self-triggered thermoelectric catalysis based on BST/CaO₂ heterojunction combined catalytic therapy, ion

interference therapy, and immunotherapy exhibited excellent antitumor and biosafety properties both *in vitro* and *in vivo*.

Although the self-triggered synergistic thermoelectric, ionic interference, and immunotherapy demonstrated in this study show promising advantages and potential applications, further research is needed before clinical use. For instance, more detailed and comprehensive analysis of material metabolic pathways and toxicological implications should be conducted. Additionally, the thermoelectric materials based on BST NSs used in this study degrade slowly *in vivo*, leading to accumulation in vital organs. Although no adverse reactions were detected during the short-term study, it is challenging to predict long-term residual toxicity *in vivo*. Therefore, developing new safe and efficient degradable thermoelectric materials coupled with tumor-specific switches, such as hypoxia-responsive, low pH-responsive, and high ROS-responsive materials, would be a crucial strategy to promote the clinical transformation of self-triggered thermoelectric therapy.

10. In Fig. 1, what do the gray ovals on cell membrane mean?

Response: We apologize for this careless omission. The gray ovals on the cell membrane indicate calcium channel proteins, which have been added to our revised manuscript.

Fig. 1. Schematic diagram of the synthesis of BST/CaO₂ NSs and the mechanism of self-triggered thermoelectric therapy for cancer treatment.

11. The manuscript contains many not-so-frequently used expressions, it would benefit from a thorough review for grammatical errors and typos.

Response: Thank you for the comments. We have revised the **WHOLE** manuscript carefully and tried to avoid any grammar or syntax errors. In addition, we have asked colleagues who are skilled in writing scientific papers in English to check the English. We believe that the language is much improved.

Reviewer #3 (Remarks to the Author):

This paper describes an interesting thermal triggered nanocatalyst system that generate ROS in acidic TME. Both *in vitro* and *in vivo* data support the multifunctional effects of BST/CaO₂ NSs in generating Ca²⁺ ion surges, heat, and ROS. However, some critical control experiments are missing. Background information and literature on prior work, especially prior work on CaO₂, is completely omitted. BST and CaO₂ nanosystems have been separately reported before.

Response: We very much appreciate the reviewer's thoughtful and helpful comments. During the past two months, we have performed a series of additional experiments to acquire more significant data. All these data have been added accordingly to the revised manuscript. Moreover, we have also made a series of modifications/corrections/additions to the manuscript. We hope that this revised version can now address all the concerns raised by the respected reviewer and satisfy the high publication standard in *Nature Communications*. Below, please also find our point-by-point responses.

For the *innovation* of our strategy reported in this manuscript, although both BST NSs and CaO₂ NPs have been reported separately for tumor therapy, the intelligent combination of BST NSs and CaO₂ NPs and their synergy-derived tumor-specific self-triggered thermoelectric therapy is what makes this paper innovative. The exogenous excitation requirement and electron-hole pair recombination are the key elements limiting the application of catalytic therapies. Tumor-specific self-triggered thermoelectric catalysis based on BST/CaO₂ heterojunction combined catalytic therapy, ion interference therapy, and immunotherapy is first reported.

Central innovation:

Tumor microenvironment (TME)-specific and self-triggered thermoelectric therapy without any external stimulation is innovative and has great potential in tumor therapy.

Collaborative innovation:

(1) Upon exposure to the acidic TME, the CaO₂ NP coating hydrolyzed rapidly and released Ca²⁺, H₂O₂, and heat.

(2) Heat: The heat induced a temperature difference on BST NSs, triggering the thermoelectric effect, which pyro-generates negative and positive charges for chemical oxidation–reduction reactions and reactive oxygen species (ROS) generation. The voltage inside the thermoelectric material (BST NSs)-induced self-built-in electric field can retard electron-hole recombination, ensuring the corresponding catalytic activity and high ROS production.

(3) H₂O₂: H₂O₂ not only provides substrate (O₂) supplementation for thermoelectric catalysis but also dysregulates Ca²⁺ channels, preventing Ca²⁺ efflux.

(4) Ca²⁺: Ca²⁺ mediates calcium overload-mediated therapy, which could be aggravated by dysregulation of calcium channels by H₂O₂. Additionally, Ca²⁺ promotes DC maturation and tumor antigen presentation, thus activating the immune response and enabling immunotherapy.

Biosafety:

In a normal physiological environment, the different and mild hydrolysis pathways of CaO₂ NPs, producing Ca²⁺ and O₂ slowly and without heat, cannot trigger the thermoelectric catalysis of BST NSs, guaranteeing high biosafety to normal organs and tissues.

The innovation of our reported TME-specific and self-triggered thermoelectric therapy was reframed and emphasized in the Abstract, Introduction, and Discussion sections.

Control experiments needed for critical evaluation of the reported work:

a. Synthesis and characterization of CaO₂ NPs.

Response: Thank you for the reviewer's comments. The synthesis and characterization of CaO₂ NPs have been added to our revised manuscript.

Methods section:

Preparation of CaO₂ NPs.

CaCl₂ (0.1 g) and PVP (0.35 g) were weighed into a round flask and dissolved in 15 mL of ethanol using an ultrasound device. While stirring, 1 mL of ammonia and 0.2 mL of H₂O₂ solution were slowly added to the mixture to obtain a light blue milky white solution. The resulting product was collected by centrifugation at 15,000 rpm, washed three times with ethanol, and finally redispersed in deionized water.

Results section:

CaO₂ NPs were synthesized using CaCl₂, ammonia, and H₂O₂ as substrates in an ethanol solution. Both SEM and TEM images of CaO₂ NPs showed that the synthesized CaO₂ NPs had a uniform morphology with an average size of 10 nm (Fig. 2b and 2f). EDS mapping of CaO₂ NPs also confirmed successful preparation (Fig. 2j, Supplementary Fig. 2).

Fig. 2. Characterization of prepared BST NSs, CaO₂ NPs, and BST/CaO₂ NSs. **a** SEM images of BST NSs. Scale bar = 50 nm. **b** SEM images of CaO₂ NPs. Scale bar = 50 nm. **c, d** SEM images of BST/CaO₂ NSs. Scale bar = 50 nm for **c**, Scale bar = 10 nm for **d**. **e** TEM images of BST NSs. Scale bar = 100 nm. **f** TEM images of CaO₂ NPs. Scale bar = 100 nm. **g, h** TEM images of BST/CaO₂ NSs. Scale bar = 100 nm for **g**, Scale bar = 5 nm for **h**. **i** SEM image and elemental mappings of BST NSs, including Bi, Sb, and Te elements. Scale bar = 100 nm. **j** SEM image and elemental mappings of CaO₂ NPs. Scale bar = 100 nm. **k** BST/CaO₂ NSs, including Bi, Sb, Te, Ca and O elements. Scale bar = 100 nm.

b. Cell viability and cell death at acidic pH.

Response: Thank you for the reviewer's comments. Acidic pH due to high-efficiency glycolysis is one of the most representative specificities of tumor cells. Hence, tumor cells have good viability at acidic pH.

Fig. 5. Cell viability of BST NSs, CaO₂ NPs and BST/CaO₂ NS-treated **a** CT26 cells and **b** TE1 cells by CCK8 assays. Data are presented as the mean \pm s.d. ($n = 5$ biologically independent cells). Statistical differences were analyzed by Student's two-sided t test.

c. Temperature change (ΔT) in culture media for CaO₂, BST, and BST/CaO₂.

Response: Thank you for the reviewer's comments. Temperature changes (ΔT) in culture media with different pH values (7.4 and 5.5) for CaO₂, BST, and BST/CaO₂ were detected and added to our revised manuscript.

Results section:

The temperature difference of the thermoelectric catalyst is an indispensable condition for triggering the thermoelectric effect. Therefore, in this study, CaO₂ was assembled onto the surface of BST NSs *in situ* to create BST/CaO₂ NSs. The trigger (CaO₂) could only be activated by the low pH of the tumor microenvironment (TME), which triggered the thermoelectric effect. To test the theory of the low pH-specific self-triggered thermoelectric effect, the temperature change of BST NSs, CaO₂ NPs, and BST/CaO₂ NSs at different pH values was detected. It was observed that there was a rapid temperature rise when CaO₂ NPs or BST/CaO₂ NSs were placed in a low pH solution (pH 5.5), but there was no significant temperature fluctuation at neutral pH (pH 7.4) (Fig. 3e).

Fig. 3. Analysis of the catalytic performance of the self-triggered thermoelectric system. **a** Zeta potential of ligand-free BST, BST-PAA-Ca²⁺, and BST/CaO₂ NSs. Data are presented as the mean ± s.d. (n = 3 independent experiments). **b** X-ray diffraction (XRD) patterns of BST NSs and BST/CaO₂ NSs. **c** X-ray photoelectron spectroscopy (XPS) spectra of BST NSs, CaO₂ NPs, and BST/CaO₂ NSs. **d** High-resolution XPS spectra of BST/CaO₂ NSs (Bi 4f, Sb 3d, Te 3d, Ca 2p and O 1 s). **e** The temperature change of BST NSs, CaO₂ NPs, and BST/CaO₂ NSs at different pH values. Degradation of DPBF by **f** BST and **g** BST/CaO₂ NSs at pH 5.5. **h** Reaction mechanism of DPBF detection- $\cdot\text{O}_2^-$. **i** Degradation of DPBF by different groups. Data are presented as the mean ± s.d. (n = 3 independent experiments).

d. Intracellular concentration of Ca²⁺ ions.

Response: Thank you for the comments. Intracellular concentrations of Ca²⁺ ions

before and after treatment with CaO₂, BST, and BST/CaO₂ have been tested and added to our revised manuscript.

Results section:

To validate the efficacy of calcium overload-mediated ion interference therapy, the intracellular concentrations of Ca²⁺ were evaluated both qualitatively and quantitatively using confocal laser scanning microscopy (CLSM) and flow cytometry (FCM) with a Fluo-4 AM probe. The results depicted in Supplementary Fig. 10a indicate a significant increase in green fluorescence after treatment with CaO₂ NPs and BST/CaO₂ NSs, indicating successful endocytosis of these particles by tumor cells and hydrolysis of CaO₂ at low pH. Additionally, the quantitative data on intracellular Ca²⁺ concentrations obtained *via* FCM analysis (Supplementary Fig. 10b) confirmed the high level of endocytosis observed for CaO₂ NPs and BST/CaO₂ NSs and the low pH responsive hydrolysis of CaO₂ in tumor cells.

Supplementary Figure 10. Analysis of intracellular Ca²⁺ concentrations of different treatments. a Confocal laser scanning microscopy (CLSM) images and b flow cytometer (FCM) analysis of intracellular Ca²⁺ concentrations of different treatments. Scale bar = 10 μ m.

Several minor points need to be addressed.

1. Figure 2: what are the values for scale bars?

Response: We apologize for this careless omission. The values for scale bars have been added in our revised manuscript.

Fig. 5. *In vitro* antitumor performance of the self-triggered thermoelectric system. Cell viability of BST NSs, CaO_2 NPs and BST/ CaO_2 NS-treated **a** CT26 cells and **b** TE1 cells by CCK8 assays. Data are presented as the mean \pm s.d. ($n = 5$ biologically

independent cells). Statistical differences were analyzed by Student's two-sided t test. Representative fluorescence images and quantification of intracellular ROS by **c** CLSM and **d** FCM. Scale bar = 100 μm . **e** Representative confocal microscopy images of mitochondria-selective JC-1-stained CT26 cells after different treatments. Scale bar = 10 μm . **f** Representative confocal microscopy images of γ -H2AX-stained CT26 cells after different treatments. Scale bar = 10 μm . **g** Confocal imaging of CT26 cells stained with PI (red fluorescence) and Calcein-AM (green fluorescence) to distinguish dead cells and live cells after different treatments. Scale bar = 100 μm . **h** FCM images of CT26 cells stained with PI (red fluorescence) and Annexin V-FITC (green fluorescence) to measure cell apoptosis after treatment under different conditions.

2. Evidence of heat generated when BST/CaO₂ NSs are exposed to aqueous media at different pH values.

Response: Thank you for the reviewer's comments. Temperature changes (ΔT) in culture media with different pH values (7.4 and 5.5) for CaO₂, BST, and BST/CaO₂ were detected and added to our revised manuscript.

Results section:

The temperature difference of the thermoelectric catalyst is an indispensable condition for triggering the thermoelectric effect. Therefore, in this study, CaO₂ was assembled onto the surface of BST NSs *in situ* to create BST/CaO₂ NSs. The trigger (CaO₂) could only be activated by the low pH of the tumor microenvironment (TME), which triggered the thermoelectric effect. To test the theory of the low pH-specific self-triggered thermoelectric effect, the temperature change of BST NSs, CaO₂ NPs, and BST/CaO₂ NSs at different pH values was detected. It was observed that there was a rapid temperature rise when CaO₂ NPs or BST/CaO₂ NSs were placed in a low pH solution (pH 5.5), but there was no significant temperature fluctuation at neutral pH (pH 7.4) (Fig. 3e).

Fig. 3. Analysis of the catalytic performance of the self-triggered thermoelectric system. **a** Zeta potential of ligand-free BST, BST-PAA-Ca²⁺, and BST/CaO₂ NSs. Data are presented as the mean ± s.d. (n = 3 independent experiments). **b** X-ray diffraction (XRD) patterns of BST NSs and BST/CaO₂ NSs. **c** X-ray photoelectron spectroscopy (XPS) spectra of BST NSs, CaO₂ NPs, and BST/CaO₂ NSs. **d** High-resolution XPS spectra of BST/CaO₂ NSs (Bi 4f, Sb 3d, Te 3d, Ca 2p and O 1 s). **e** The temperature change of BST NSs, CaO₂ NPs, and BST/CaO₂ NSs at different pH values. Degradation of DPBF by **f** BST and **g** BST/CaO₂ NSs at pH 5.5. **h** Reaction mechanism of DPBF detection·O₂⁻. **i** Degradation of DPBF by different groups. Data are presented as the mean ± s.d. (n = 3 independent experiments).

3. Figure 5: what were injected dose for each nanoparticle? What were the concentrations of Ca^{2+} in tumor and lymph nodes?

Response: Thank you for the helpful comments from the reviewer. The injected dose of each treatment has been added to our Methods section. The concentrations of Ca^{2+} in the tumor and lymph nodes were also detected and added to our revised manuscript.

Methods section:

Measurement of antitumor effects.

Subcutaneous xenograft colorectal cancer animal model:

The CT26 tumor-bearing mice were randomly divided into four treatment groups, and the tumors reached approximately 80 mm^3 with five mice each as follows: PBS, BST NSs, CaO_2 NPs, and BST/ CaO_2 NSs. The BST NSs and BST/ CaO_2 NSs were injected intravenously at a dose of 5 mg/kg. Because the loading capacity of CaO_2 NPs on BST/ CaO_2 NSs was 10 wt%, the CaO_2 NPs were injected intravenously at a dose of 0.5 mg/kg. The body weight and tumor size of each mouse in the different groups were measured and recorded by a caliper and digital scale every 2 days during the treatment. The tumor volumes were calculated according to the following formula: tumor volume = $(\text{length} \times \text{width}^2)/2$.

Results section:

To further confirm the Ca^{2+} -mediated ion interference therapy, the concentration of Ca^{2+} in tumor cells was measured. The results demonstrated that each injection increased the Ca^{2+} concentration in the subcutaneous graft tumor cells, which gradually returned to normal levels over time (Supplementary Fig. 15).

To confirm the role of Ca^{2+} in promoting the maturation of DC cells, we measured the concentration of Ca^{2+} in lymphocytes at lymph nodes. As depicted in Supplementary Fig. 15, the concentration of Ca^{2+} in lymphocytes was positively correlated with the number of injections, indicating that an increase in Ca^{2+} concentration could be detected in lymphocytes 12 hours after each injection. However, over time, the concentration of Ca^{2+} in the lymphocytes gradually returned to normal levels.

Although three doses of BST/ CaO_2 NSs were injected into each mouse, the concentrations of both BST NSs and CaO_2 NPs were very low, at 0.1 and 0.01 mg per mouse, respectively. The concentrations of Ca^{2+} in blood exhibited no significant fluctuation due to the slow hydrolysis of CaO_2 NPs under normal conditions (Supplementary Fig. 15).

Supplementary Figure 15. The concentrations of Ca^{2+} in blood, tumor cells, and lymphocytes at different times under treatment with BST/ CaO_2 NSs. The black arrow represents the three injection times.

Reviewers' Comments:

Reviewer #1:

Remarks to the Author:

The authors fully addressed my comments and suggestions. My recommendation is to accept as it is.

Reviewer #2:

Remarks to the Author:

The authors have responded to the concerns raised by the reviewers and have performed additional experiments to address all comments. They have included in vivo fluorescence images of three mice in Figure 6 and have performed PA and CT imaging for more accurate characterization of the distribution of nanomaterials in vivo. Relevant statements regarding these additions have been included in the revised manuscript.

- Innovation and concept concerns:

The authors have concluded that the major innovation of the study is the intelligent combination of BST NSs and CaO₂ NPs, which leads to tumor-specific self-triggered thermoelectric therapy without any external stimulation.

1. The concept "thermoelectric therapy" has specific meaning in clinical medicine, where a semiconductor is used to generate precisely controlled heating and cooling for medical purposes. The therapeutic effects of BST/CaO₂ heterojunction nanoparticles, according to the results in this study, are quite complex, including heat release (CaO₂ at low pH), ROS generation (dynamic therapy), calcium ion overload, and anti-cancer immune activation resulted from Ca-related DC maturation.

2. When claiming the thermoelectric therapeutic effect of BST/CaO₂, the heating and cooling control is not measured and the reviewer is not sure how could this be tested for a nano-sized material.

3. Heat-generated ROS: According to Fig. 3e, CaO₂ can generate heat under acidic environment, then triggers the generation of ROS. Also on Page 10, the authors stated that "The ability of BST/CaO₂ NSs to induce tumor cell death is thought to be mainly due to the production of ROS". Does this mean that heat/thermo-dynamic therapy is the key to the treatment, instead of the so-called "thermoelectric therapy" in the title.

- Technical concerns:

4. It looks like CaO₂ can also elicit treatment effect not only on the cellular level, but in animals too, especially considering the fact that the administrated dose of CaO₂ (0.5 mg/kg) is only 10% of BST/CaO₂ (5 mg/kg).

5. No method description was found regarding Fig. 3e;

6. Is there any reason that the nanoparticle would go to the lung? Would they aggregate when injected into the blood and a portion was stuck in the lung?

7. 24 h is a bit short for toxicity evaluation, given its multiple injection treatment plan and significant background uptake in the liver, lung, and spleen.

- Animal welfare concerns:

8. According to the "Guidelines for the welfare and use of animals in cancer research - P. Workman et al. - British Journal of Cancer (2010):" For an animal carrying a single tumour, the mean diameter should not normally exceed 1.2 cm in mice or 2.5 cm in rats, or 1.5 and 2.8 cm, respectively, for therapeutic studies." The end-point used in this study is 3,000 cubic millimeter, exceeds the limit mentioned in the guideline.

- Figure concerns:

9. Fig. 5c, the images of the Control group and CaO₂ group look a bit similar;

10. Fig. 6f should have a scale bar showing the HU range of CT images;

Reviewer #3:

Remarks to the Author:

My questions and critiques have been adequately addressed. The paper is now acceptable.

Reviewer #1 (Remarks to the Author):

The authors fully addressed my comments and suggestions. My recommendation is to accept as it is.

Response: Thank you very much for your kind words after revision.

Reviewer #2 (Remarks to the Author):

The authors have responded to the concerns raised by the reviewers and have performed additional experiments to address all comments. They have included *in vivo* fluorescence images of three mice in Figure 6 and have performed PA and CT imaging for more accurate characterization of the distribution of nanomaterials *in vivo*. Relevant statements regarding these additions have been included in the revised manuscript.

Response: We very much appreciate the reviewer's thoughtful and helpful comments. The comments and suggestions of reviewers have provided great help to the integrity and depth of our research.

- Innovation and concept concerns:

The authors have concluded that the major innovation of the study is the intelligent combination of BST NSs and CaO₂ NPs, which leads to tumor-specific self-triggered thermoelectric therapy without any external stimulation.

Response: Thank you very much for your kind words and recognition of the innovation of this research.

1. The concept “thermoelectric therapy” has specific meaning in clinical medicine, where a semiconductor is used to generate precisely controlled heating and cooling for medical purposes. The therapeutic effects of BST/CaO₂ heterojunction nanoparticles, according to the results in this study, are quite complex, including heat release (CaO₂ at low pH), ROS generation (dynamic therapy), calcium ion overload, and anti-cancer immune activation resulted from Ca-related DC maturation.

Response: Thank you for this valuable comment. The main reason why we call it “thermoelectric therapy” is that the core mechanism of this novel therapy is that the thermoelectric effect of the BST NSs mediates electron-hole separation, thereby triggering subsequent catalytic reactions (ROS generation). The hydrolysis of the CaO₂ NP coating in TME releases heat is the trigger of the thermoelectric effect of the BST NSs. Other synergies, including calcium ion overload and Ca-related DC maturation, are additional effects of this system.

As you said, the concept “thermoelectric therapy” has specific meaning in clinical medicine. To be more rigorous, the thermoelectric effect-based therapy reported in this study was changed to “thermoelectric dynamic therapy”.

2. When claiming the thermoelectric therapeutic effect of BST/CaO₂, the heating and cooling control is not measured and the reviewer is not sure how could this be tested for a nano-sized material.

Response: Thank you for this valuable comment. As mentioned in Question 1, this BST/CaO₂ heterojunction-mediated cancer therapy should be called “thermoelectric dynamic therapy”, in which ROS generation is a major antitumor principle. Many studies have confirmed that BST is an excellent thermoelectric material, but as the

reviewer said, there is truly no way to detect the heating and cooling control of nanomaterials.

3. Heat-generated ROS: According to Fig. 3e, CaO₂ can generate heat under acidic environment, then triggers the generation of ROS. Also on Page 10, the authors stated that “The ability of BST/CaO₂ NSs to induce tumor cell death is thought to be mainly due to the production of ROS”. Does this mean that heat/thermo-dynamic therapy is the key to the treatment, instead of the so-called “thermoelectric therapy” in the title.

Response: Thank you for this valuable comment. The concept “thermo-dynamic therapy” also has specific meaning in clinical medicine. It uses heat as an energy source to activate sensitizers and produce reactive chemical species, including ROS and oxygen-irrelevant free radicals, for cancer therapy. The main principle of thermodynamic therapy is the use of a small number of heat-resistant azobi (isobutyronitrile) derivatives that are easily decomposed to produce free radicals during thermal activation. Hence, as we explained in Questions 1 and 2, “thermoelectric dynamic therapy” should be more suitable to define the antitumor approach reported in this study. We have revised the whole manuscript regarding the definition of BST/CaO₂ heterojunction-mediated cancer therapy.

- Technical concerns:

4. It looks like CaO₂ can also elicit treatment effect not only on the cellular level, but in animals too, especially considering the fact that the administrated dose of CaO₂ (0.5 mg/kg) is only 10% of BST/CaO₂ (5 mg/kg).

Response: Thank you for this valuable comment. Calcium overload, characterized by an abnormal cytoplasmic accumulation of free calcium ions (Ca²⁺), is a widely recognized cause of damage in numerous cell types and even of cell death. This undesirable destructive process has become a new tool applicable to cancer treatment. In our previous review article, we presented a comprehensive overview of recent research works on Ca²⁺-based nanosystems for tumor therapy (*Coordination Chemistry Reviews* 481 (2023) 215050). The principles of Ca²⁺-based tumor therapy are summarized from two main aspects: regulation of immune cells and direct induction of tumor cell death.

In this research, although CaO₂ NPs are not the focus of this study, they also play an important role in the excellent curative effect. Upon exposure to the acidic TME, the CaO₂ NP coating hydrolyzed rapidly and released Ca²⁺, H₂O₂, and heat. Each element can cause damage to tumor cells, and at the same time, when combined with BST NSs, these elements produce greater antitumor effects, greatly enhancing the synergy of this therapy.

CaO₂ NPs, or other Ca²⁺-based nanosystems, have become a new tool applicable to cancer treatment. Many researchers, including us, will invest more effort in follow-up research on calcium-mediated tumor therapy. We look forward to sharing our findings with you in the near future.

5. No method description was found regarding Fig. 3e;

Response: Thank you for your helpful comments. The method description regarding Fig. 3e has been added in our revised manuscript.

Methods section:

Temperature changes at different pH values.

To determine the trigger effect of CaO₂ NPs, the temperature change of BST NS, CaO₂ NP and BST/CaO₂ NS solutions was recorded. In detail, BST NSs, CaO₂ NPs and BST/CaO₂ NSs were prepared and added to PBS solutions (final concentration 0.1 mg/mL) with different pH values (pH 7.4 and pH 5.5). The temperature of the solution was detected and recorded by an infrared thermometer.

6. Is there any reason that the nanoparticle would go to the lung? Would they aggregate when injected into the blood and a portion was stuck in the lung?

Response: Thank you for your comments. The accumulation of nanomaterials in the lung has been explained and added to our revised manuscript.

Results section:

As shown by fluorescence and CT imaging *in vivo*, nanomaterials not only accumulate in tumors but also accumulate in the liver, kidney, spleen and lung. Because the liver, kidney and spleen are the main metabolic organs, nanomaterials are mainly excreted through the metabolism of the above three organs. The accumulation in the lungs is mainly due to the following reasons. The lung is a highly vascularized organ with a rich network of capillaries. The small size of the nanomaterials allows them to penetrate into lung tissue through the walls of blood vessels. The short distance between the alveoli and the pulmonary capillaries also promotes the deposition and aggregation of nanomaterials in the lungs.^{58,59} In addition, the lung contains a large number of alveolar macrophages, which participate in the absorption and metabolism of foreign molecules and particles.⁶⁰ When the nanomaterials are injected intravenously, various serum proteins bind to the nanomaterials and are recognized and internalized by scavenger receptors on the surface of the macrophages, resulting in the aggregation of the nanomaterials in the lungs. However, it is eventually cleared out of the body by macrophages.⁶¹ Moreover, to further investigate the biodistribution of BST/CaO₂ NSs *in vivo*, ICP/MS analysis was utilized, as depicted in Supplementary Fig. 14. The results indicated a significant accumulation of NSs within the major organs and tumors over a period of 30 days, highlighting their effectiveness in targeting tumors. Importantly, Supplementary Fig. 14 also illustrates that the accumulated BST/CaO₂ NSs within normal organs and tissues were gradually excreted by the body over time, indicating their biocompatibility and potential for clinical translation.

58. Yang, S. T., Luo, J., Zhou, Q., Wang, H. Pharmacokinetics, metabolism and toxicity of carbon nanotubes for biomedical purposes. *Theranostics* **2**, 271-282 (2012).
59. Wang, H., Yang, S.-T., Cao, A., Liu, Y. Quantification of Carbon Nanomaterials *in Vivo*. *Acc. Chem. Res.* **46**, 750-760 (2013).
60. Huang, X., Li, L., Liu, T., Hao, N., Liu, H., Chen, D., Tang, F. The Shape Effect of Mesoporous Silica Nanoparticles on Biodistribution, Clearance, and Biocompatibility *in Vivo*. *ACS Nano* **5**, 5390-5399 (2011).
61. Fujihara, J., Tongu, M., Hashimoto, H., Yamada, T., Kimura-Kataoka, K., Yasuda, T., Fujita, Y., Takeshita, H. Distribution and toxicity evaluation of ZnO dispersion nanoparticles in single intravenously exposed mice. *J. Med. Invest.* **62**, 45-50 (2015).

7.24 h is a bit short for toxicity evaluation, given its multiple injection treatment plan and significant background uptake in the liver, lung, and spleen.

Response: We appreciate this helpful comment. In the meantime, we apologize for our oversight. In fact, the apoptosis, DNA damage, HE staining, and inflammatory factor analysis exhibited in Figure 9, Figure 10a and 10b were carried out on the tenth day of different treatments. The blood hematology and blood biochemical analysis exhibited in Figure 10c and 10d were carried out after 1 day, 7 days, and 14 days of injections. The corresponding statement has been revised.

Methods section:

***In vivo* biosafety measurements.**

To investigate the biosafety of BST/CaO₂ NS-based cancer therapy, the apoptosis and DNA damage of vital organs (including heart, liver, spleen, lung, and kidney) of subcutaneous xenograft colorectal cancer mice after different treatment treatments were investigated. On the tenth day of treatment, the mice were euthanized, and their vital organs and tumors were extracted for flow cytometry and H&E staining analysis. For the BST/CaO₂ NS-treated mice, inflammatory factors, including TNF- α , IL-8, IL-1 β , IL-6, and IFN- γ , were detected by real-time quantitative PCR (RT-qPCR) on the tenth day of treatment.

In addition, healthy BALB/c mice (female, 7 weeks, 16–18 g) were intravenously injected with BST/CaO₂ NSs at a high dose of 10 mg/kg. At 1 day, 7 days, and 14 days after injection, complete blood counts including platelet (PLT), white blood cells (WBC), mean corpuscular volume (MCV), red blood cells (RBC), mean corpuscular hemoglobin concentration (MCHC), mean corpuscular hemoglobin (MCH), hematocrit (HCT), hemoglobin (HGB), and serum biochemical parameters including γ -glutamyl transpeptidase (γ -GT), total protein (TP), creatine kinase (CK), lactate dehydrogenase (LDH), c-reactive protein (CRP), creatinine (Cr), blood urea nitrogen (BUN), alanine aspartate aminotransferase (AST), amylase (AMY), aminotransferase (ALT), and albumin (ALB) were detected and compared with the control group to estimate the biocompatibility of BST/CaO₂ NSs.

- Animal welfare concerns:

8. According to the “Guidelines for the welfare and use of animals in cancer research - P. Workman et al. - British Journal of Cancer (2010):“ For an animal carrying a single tumour, the mean diameter should not normally exceed 1.2 cm in mice or 2.5 cm in rats, or 1.5 and 2.8 cm, respectively, for therapeutic studies.” The end-point used in this study is 3,000 cubic millimeter, exceeds the limit mentioned in the guideline.

Response: We appreciate the reviewer’s thoughtful and helpful comments.

In this study, the endpoint of tumor-bearing mice was set at 2000 cm³. The tumor volume of mice was measured every two days, so although the tumor volume of mice in the control group exceeded 2000 cm³ on the tenth day, the volume of most mice was below 2000 cm³ on the eighth day. In addition, we strictly followed the Guidelines for the Care and Use of Laboratory Animals and closely monitored the survival status of the tumor-bearing mice. Even on the tenth day, the tumor-bearing mice lived well, with regular diet and activity, stable weight, and no obvious pain. As this study involves immunotherapy, we intend to observe the metastasis of tumor-bearing mice in the advanced stage of the tumor. Therefore, under the condition of ensuring the mice were in good condition, we euthanized tumor-bearing mice and dissected them for tumor metastasis analysis on the tenth day. We will further investigate the role of this therapy in inhibiting tumor metastasis in follow-up studies. In addition, lots of biosafety analysis, including apoptosis, DNA damage, HE staining and inflammatory factors of

major organs, were carried out on the tenth day of treatment. All animal experimental protocols were approved by the Animal Ethics Committee. Moreover, we have clarified the exceptions of tumor burden in the Methods section.

Methods section:

Establishment of subcutaneous xenograft colorectal cancer animal model.

All animal experiments were approved by the Animal Ethics Committee of the Tianjin University Laboratory Animal Center (Tianjin, China) and carried out according to the Guidelines for the Care and Use of Laboratory Animals of Tianjin University. 2000 mm³ was set as the permitted maximal tumor size in our *in vivo* antitumor experiments. One hundred microliters of serum-free cell medium containing 2×10^6 CT26 cells was injected subcutaneously into BALB/c mice (female, 6 weeks, 14-16 g) to establish a xenograft tumor model.

Subcutaneous xenograft colorectal cancer animal model:

The CT26 tumor-bearing mice were randomly divided into four treatment groups, and the tumors reached approximately 80 mm³ with five mice each as follows: PBS, BST NSs, CaO₂ NPs, and BST/CaO₂ NSs. The BST NSs and BST/CaO₂ NSs were injected intravenously at a dose of 5 mg/kg. Because the loading capacity of CaO₂ NPs on BST/CaO₂ NSs was 10 wt%, the CaO₂ NPs were injected intravenously at a dose of 0.5 mg/kg. The tumor size of each mouse in the different groups was measured and recorded by a caliper and digital scale every 2 days during the treatment. The tumor volumes were calculated according to the following formula: tumor volume = (length × width²)/2. To further study the metastasis of advanced colorectal cancer and biosafety of different treatments, five mice in each group were euthanized on the tenth day under the premise of ensuring a regular diet, stable weight and no obvious pain. All animal experimental protocols were approved by the Animal Ethics Committee.

- Figure concerns:

9.Fig. 5c, the images of the Control group and CaO₂ group look a bit similar;

Response: We appreciate this helpful comment. We apologize for this mistake. Since the folders containing the fluorescent photos of the control group and CaO₂ group were both named with the capital letter C, the data were processed without carefully looking at the full names of the folders, which led to confusion between the two sets of photos. The relevant picture has been replaced with the correct picture. We have rechecked all the original data, and the other data in this article are correct.

Fig. 5. *In vitro* antitumor performance of the self-triggered thermoelectric system. Cell viability of BST NSs, CaO₂ NPs and BST/CaO₂ NS-treated **a** CT26 cells and **b** TE1 cells by CCK8 assays. Data are presented as the mean \pm s.d. (n = 5 biologically independent cells). Statistical differences were analyzed by Student's two-sided t test. Representative fluorescence images and quantification of intracellular ROS by **c** CLSM and **d** FCM. Scale bar = 100 nm. **e** Representative confocal microscopy images of mitochondria-selective JC-1-stained CT26 cells after different treatments. Scale bar = 10 nm. **f** Representative confocal microscopy images of γ -H2AX-stained CT26 cells

after different treatments. Scale bar = 10 mm. **g** Confocal imaging of CT26 cells stained with PI (red fluorescence) and Calcein-AM (green fluorescence) to distinguish dead cells and live cells after different treatments. Scale bar = 100 mm. **h** FCM images of CT26 cells stained with PI (red fluorescence) and Annexin V-FITC (green fluorescence) to measure cell apoptosis after treatment under different conditions.

10. Fig. 6f should have a scale bar showing the HU range of CT images;

Response: Thank you for this comment. The scale bar showing the HU range of CT images has been added in our revised manuscript.

Fig. 6. *In vivo* imaging and biodistribution of the BST/CaO₂ heterojunction. **a** *In vivo* fluorescence images of tumor-bearing mice at different time points after intravenous injection with Cy5.5-labeled BST/CaO₂ heterojunction and **b** *ex vivo* fluorescence images of tumor and major organs at 24 hours after injection. **c** *In vivo* photoacoustic images of tumor-bearing mice after intravenous injection with BST/CaO₂ heterojunction. **d** CT images of BST/CaO₂ heterojunctions with different concentrations. **e** The CT values (HU) of the BST/CaO₂ heterojunction. **f** *In vivo* CT

images of tumor-bearing mice after intravenous injection with the BST/CaO₂ heterojunction. The red circle indicates a tumor.

Reviewer #3 (Remarks to the Author):

My questions and critiques have been adequately addressed. The paper is now acceptable.

Response: Thank you very much for your kind words after revision.

Reviewers' Comments:

Reviewer #2:

Remarks to the Author:

The authors have adequately addressed all my concerns and the reviewer supports the publication of the revised manuscript as it is.

REVIEWERS' COMMENTS:

Reviewer #2 (Remarks to the Author):

The authors have adequately addressed all my concerns and the reviewer supports the publication of the revised manuscript as it is.

Response: Thank you very much for your kind words after revision.